# Coupling Generative Modeling and an Autoencoder with the Causal Bridge

**Ruolin Meng   Ming-Yu Chung   Dhanajit Brahma   Ricardo Henao   Lawrence Carin**
Duke University
{ruolin.meng,ming-yu.chung,dhanajit.brahma,r.henao,lcarin}@duke.edu

## Abstract

We consider inferring the causal effect of a treatment (intervention) on an outcome of interest in situations where there is potentially an unobserved confounder influencing both the treatment and the outcome. This is achievable by assuming access to two separate sets of control (proxy) measurements associated with treatment and outcomes, which are used to estimate treatment effects through a function termed the *causal bridge* (CB). We present a new theoretical perspective, associated assumptions for when estimating treatment effects with the CB is feasible, and a bound on the average error of the treatment effect when the CB assumptions are violated. From this new perspective, we then demonstrate how coupling the CB with an autoencoder architecture allows for the sharing of statistical strength between observed quantities (proxies, treatment, and outcomes), thus improving the quality of the CB estimates. Experiments on synthetic and real-world data demonstrate the effectiveness of the proposed approach relative to state-of-the-art methodology for causal inference with proxy measurements.

## 1 Introduction

Estimating the causal effect of a treatment on an outcome is crucial in various domains, but the presence of unobserved confounders can hinder accurate inference [36, 44]. Traditional methods often rely on strong assumptions, such as the absence of unobserved confounders [43, 40]. Other approaches have assumed available instrumental variables with specific properties [2, 51, 19]. However, these assumptions may not always hold in practice, leading to biased estimates of causal effects.

A promising approach to address this challenge is the use of *proxy variables* [17, 30, 28], also known as *negative control variables*. These are variables that are affected by the unobserved confounder, but do not necessarily directly influence the treatment or outcome. Consequently, leveraging information from these proxies, we can gain insight into the underlying causal mechanisms and potentially mitigate the bias caused by unobserved confounders.

Recent work has introduced the concept of a *causal bridge function*, which uses two sets of proxy variables, one related to the treatment and the other to the outcome, to estimate causal effects [30, 31, 52]. This approach has shown promising results, but there is still room for improvement in terms of both theoretical understanding and practical implementation.

This paper builds on the causal bridge framework and makes several key advances. We provide a refined theoretical analysis, clarifying the assumptions and conditions under which the causal bridge function yields accurate causal effect estimates. Furthermore, we introduce a novel learning approach that leverages the power of generative models to enhance the estimation of the causal bridge. Our approach enables the sharing of statistical strength between observed variables, leading to more robust and accurate causal inference. Finally, we extend the causal bridge framework to handle survival outcomes, a common type of data in biomedical applications.

39th Conference on Neural Information Processing Systems (NeurIPS 2025).

**In summary:**

1. We develop a novel framework for causal inference with proxy variables, building upon the causal bridge function. Specifically:
   (a) We re-examine the assumptions underlying the causal bridge function, providing a new bound on the average error of the treatment effect when the assumption that the causal bridge is independent of the treatment proxy is violated (Section 4).
   (b) We introduce a new formulation for learning the causal bridge that utilizes generative models to sample from the conditional distribution of the outcome proxy given the treatment proxy and treatment (Section 6). This approach allows for more efficient and flexible estimation compared to previous methods that rely on estimating conditional expectations.
   (c) We propose an autoencoder architecture that enables the sharing of statistical strength between observed variables (Section 6), leading to improved estimation of the causal bridge and more accurate causal effect estimates.
   (d) We extend the causal bridge framework to handle survival outcomes (Section 6), broadening its applicability to important real-world problems.

2. We validate our framework through experiments on synthetic and real-world datasets (Section 7), including comparison with a randomized control trial (RCT). Our results illustrate the implications of our assumptions and demonstrate the effectiveness of our approach compared to state-of-the-art methods for causal inference with proxy variables.

## 2 Related Work

Traditional approaches for dealing with unobserved confounders often involve sensitivity analysis [16, 15], which assesses the robustness of causal effect estimates to different assumptions about the unobserved confounder [41, 39, 22, 16, 23]. Another common approach is the use of instrumental variables (IVs), variables that influence the treatment but are independent of the unobserved confounder [1, 2, 19, 51]. However, finding valid IVs can be challenging in practice.

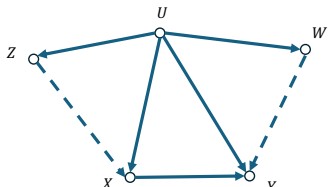

Figure 1: Graphical model for the causal-inference problem. $U$ is the unobserved confounder, $X$ is the treatment, $Y$ is the outcome of interest, and $Z$ and $W$ are the treatment and outcome controls, respectively. The dashed lines represent dependencies that may or may not be present.

More recently, there has been growing interest in using proxy variables to address unobserved confounding. Early work focused on categorical data and unobserved confounders [25], showing that causal effects can be estimated under certain conditions. The concept of causal bridge function [30, 31, 52, 24, 8] extends this idea to more general settings, using two sets of proxies to estimate causal effects. These studies have provided valuable theoretical insights and practical tools for causal inference with proxy variables.

Deep learning has also been increasingly applied to causal inference, enabling the development of more flexible models. For example, deep-generative models have been used to improve IV analysis [19] and to address unobserved confounding in various settings [28, 55, 4]. Our work builds on these advances, leveraging generative models to enhance the estimation of the causal bridge function. We note that one advantage of proxy variable methods in the unobserved confounding setting is that it requires weaker assumptions than an IV approach. Specifically, IV methods assume that the instrumental variable and the unobserved confounder are independent and the latter influences the treatment but not the outcome.

Our work draws inspiration from and contributes to several active research areas within causal inference, particularly those focused on handling unobserved confounders and leveraging proxy variables. Although our research builds upon these foundations, it makes several novel contributions, as elucidated in the previous section.

## 3 Background on the Causal Bridge

Let $X$ and $Y$ represent a treatment (intervention) and outcome, respectively. In the examples we consider, $Y$ will be continuous and $X$ could be real or categorical (with binary being an important special case). It is assumed that $X$ and $Y$ are both dependent on an *unobserved confounder $U$*, as illustrated in Figure 1. As discussed in [25, 35, 30, 31, 52], to perform inference in the presence

of the unobserved $U$, control (proxy) measurements are assumed to be available, which are also dependent on $U$. Specifically, $Z$ is a *treatment control* variable, and $W$ is an *outcome control* variable. As shown in Figure 1, the outcome may depend on $W$ and the treatment may depend on $Z$. We assume for simplicity that there are no additional covariates, but such could be included if available, as discussed in [31, 52]. We denote the domains of $(Y, X, Z, W)$ as $(\mathcal{Y}, \mathcal{X}, \mathcal{Z}, \mathcal{W})$, respectively.

For the graphical model in Figure 1, we make the following assumptions, which are consistent with those in [30, 31, 52]:

**Assumption 1 (A1)** (Latent Ignorability):

$$Y(x) \perp\!\!\!\perp X | U, \quad \forall x \in \mathcal{X}$$

**Assumption 2 (A2)** (Negative Control Outcome):

$$W \perp\!\!\!\perp X | U \text{ and } W \not\!\perp\!\!\!\perp U$$

**Assumption 3 (A3)** (Negative Control Treatment):

$$Z \perp\!\!\!\perp Y | (U, X) \text{ and } Z \perp\!\!\!\perp W | U$$

These assumptions underscore that (A1) the dependence of $X$ on $Y$ is manifest *only* through $U$; (A2) the control outcome $W$ does not *directly* influence treatment $X$; and (A3) the treatment control $Z$ does not *directly* influence the outcome $Y$ (or the control outcome $W$).

We now introduce a *bridge function* $b(W, x)$ [31, 52, 14].

**Theorem 1** [14] *If there is a solution to the Fredholm integral equation*

$$\mathbb{E}(Y | x, z) = \mathbb{E}(b(W, x) | x, z), \quad \forall \, x \in \mathcal{X} \text{ and } z \in \mathcal{Z}, \tag{1}$$

*then*

$$\mathbb{E}(Y | x, U) = \mathbb{E}[b(W, x) | U], \quad and \quad \mathbb{E}[Y | do(X = x)] = \mathbb{E}[b(W, x)].$$

The form of Theorem 1 was presented and proven in [14], and requires multiple additional assumptions to ensure that there is a solution to (1) [30, 52], and importantly, completeness, described below.

**Assumption 4 (A4)** (Completeness) *For any square-integrable function $g(\cdot)$ and for any $x$, $\mathbb{E}[g(U) | Z, x] = 0$ almost surely if and only if $g(U) = 0$ almost surely. And for any square-integrable function $h$ and for any $x$, $\mathbb{E}[h(Z) | W, x] = 0$ almost surely if and only if $h(Z) = 0$ almost surely.*

**Theorem 2** [14] *Under A1-A4, and other technical conditions, there exists a solution to* (1).

The proof and further details concerning Theorem 2 are presented in [14]. The above assumptions (and other technical requirements) are relatively abstract; therefore, in Section 4 below we explore the properties that must hold for the existence of a solution to (1). The insights from this analysis will motivate our modeling approach in Section 6.

## 4 Implied Properties of the Bridge Function

**Conditions needed for the bridge function** One may express (1) as

$$\mathbb{E}(Y | x, z) = \int dW \, b_0(W, x, z) p(W | x, z), \ b_0(W, x, z) = \int dU \, \mathbb{E}[Y | x, W, U] p(U | W, x, z). \tag{2}$$

Note that the generalized bridge function $b_0(W, x, z)$ *is a function of $z$*, *violating* the assumed form of $b(W, x)$ in (1), which is independent of $z$.

**Proposition 1** *For the equality in* (1) *to hold, either $b_0(W, x, z)$ is independent of $z$, i.e., $b_0(W, x, z) = b(W, x)$, or $b(W, x) = b_0(W, x, z) + f(W, x, z)$ and $\mathbb{E}[f(W, x, z) | x, z] = 0 \, \forall \, (x, z)$. Although $b_0(W, x, z)$ is not independent of $z$, the sum $b_0(W, x, z) + f(W, x, z)$ is.*

One possible way that $b_0(W, x, z)$ could be independent of $z$, and therefore equal to $b(W, x)$, is if $p(U | W, x, z) = p(U | W, x)$. However, we posit that $p(U | W, x, z) = p(U | W, x)$ *does not* hold in general. Another possible way for $b_0(W, x, z)$ to be equal to $b(W, x)$ is if

$$\int dU \, \mathbb{E}(Y | x, W, U) p(U | W, x, z) = \int dU \, \mathbb{E}(Y | x, W, U) p(U | W, x),$$

which implies that the $z$-dependence in $b_0(W, x, z)$ is removed after performing the expectation wrt $p(U|W, x, z)$. While this is possible, we also do not make this assumption. We therefore posit that *in general*, one cannot assume that $b_0(W, x, z)$ is independent of $z$, and therefore we conjecture that the second condition in Proposition 1 must hold, *i.e.*, $b(W, x) \neq b_0(W, x, z)$.

**Proposition 2** *If* (1) *holds, then for all* $(x, z)$, *the bridge function* $b(W, x)$ *satisfies*

$$\mathbb{E}_{W \sim p(W|x,z)}[b(W, x)] = \mathbb{E}_{W \sim p(W|x,z)}[b_0(W, x, z)], \quad (3)$$

*where* $b_0(W, x, z) = \int dU \, \mathbb{E}[Y, x, W, U] p(U|W, x, z)$. *Thus,* $b(W, x)$ *yields the correct conditional expectation of* $\mathbb{E}[Y|x, z]$ *when integrated over* $p(W|x, z)$.

The hierarchy implied from the above analysis may be summarized concisely as follows.

$$p(U|W, x, z) \neq p(U|W, x) \quad (4) \qquad \mathbb{E}[b_0(W, x, z)|x, z] = \mathbb{E}[b(W, x)|x, z] \quad (6)$$
$$b_0(W, x, z) \neq b(W, x) \quad (5) \qquad \qquad \mathbb{E}(Y|x, z) = \mathbb{E}[b_0(W, x, z)|x, z]. \quad (7)$$

Note that (7) is by definition and *is not* an assumption. Moreover, condition (6) is required for (1) to hold. This interpretation of the bridge function $b(W, x)$ as matching $b_0(W, x, z)$ *in expectation* rather than a pointwise (distributional) match is consistent with relaxed identification frameworks discussed in the literature of nonparametric instrumental variables [33, 11], where moment conditions or expectations replace the exact operator equalities.

**Bound on the Estimation Error for the Causal Bridge**  We present the following new information-theoretic result, which leverages the analysis in the previous section by relating the error in the fit to (1) with the relative information between $(U, W, Z)$. The proof is provided in Appendix B.1.

**Theorem 3** (Average Error $\eta$ for the Causal Bridge) *Assume that the function* $\mathbb{E}[Y|x, W, U]$ *is $C$-Lipschitz in $U$, and that $U$ is almost surely supported on a bounded set with* $\|U\| \leq R$. *There exists a bridge function* $b(W, x)$ *for which*

$$\mathbb{E}_{Z \sim p(Z|x)}[\underbrace{|\mathbb{E}[Y|x, Z] - \mathbb{E}_{W \sim p(W|x,Z)}[b(W, x)]|}_{\eta}] \leq CR \cdot \sqrt{2I(U; Z|W, x)}. \quad (8)$$

*One such bridge function is*

$$b(W, x) := \mathbb{E}_{U \sim p(U|W,x)}[\mathbb{E}[Y|x, W, U]],$$

*but others may exist that achieve tighter bounds.*

Examining the statement of Theorem 3, one may be tempted to suggest proxies $Z$ that are independent of $U$, but the completeness assumption (A4) concerning $\mathbb{E}[g(U)|Z, x]$ disallows it. Instead, Theorem 3 states that $W$ should be a low-noise representation of $U$, in the sense discussed in [25] concerning proxies $(W, Z)$ as "noisy" measurements of $U$.

**Corollary 1** (*W* as a noisy nonlinear mapping of *U*) *Assume that* $W = \Psi(U) + \varepsilon$, *where* $\Psi : \mathbb{R}^{d_U} \rightarrow \mathbb{R}^{d_W}$ *is an invertible, continuously differentiable* $(C^1)$ *function, and $\varepsilon$ is independent of* $(U, Z, X)$, *with zero mean and covariance matrix* $\sigma_\varepsilon^2 I$. *With additional (typical) regularity conditions provided in Appendix B.2, there exists a constant* $C_0 > 0$ *such that, for sufficiently small* $\sigma_\epsilon$

$$I(U; Z|W, x) \leq C_0 \sigma_\varepsilon^2. \quad (9)$$

The complete statement of Corollary 1 and its proof are provided in Appendix B.2. While the assumption of an invertible $\Psi(U)$ may seem strong, this same assumption was made previously for a similar setup in [25], where discrete observations and proxies were considered (see Eq. (4) in [25]).

Theorem 3 establishes that the quality of the bridge approximation critically depends on the conditional mutual information $I(U; Z|W, x)$ and Corollary 1 provides conditions under which this mutual information becomes small, namely when $W$ is a low-noise nonlinear observation of $U$ through an invertible and smooth transformation. This insight motivates practical modeling choices, namely, to construct effective bridge functions, one should design or select proxies $W$ that capture the latent confounder $U$ with minimal distortion and noise. In practice, this suggests that proxy variables with low measurement error and stable relationships to unobserved confounders are particularly valuable. Moreover, by ensuring that $W$ is an informative (albeit noisy) transformation of $U$, Corollary 1 justifies the feasibility of approximately solving the Fredholm integral equation in (1), enabling effective learning of causal bridge functions even in the presence of complex, nonlinear, confounding.

In order to illustrate the results of Theorem 3 and Corollary 1, we consider a structural equation model (SEM) for the generative process in Figure 1 (see Appendix C for details). The SEM shown in Appendix Figure 4 is consistent with the model for Corollary 1, but we now assume $\Psi(U)$ is a linear function. We make this simplification along with letting $U$ and the noise terms for $W$, $Z$, $X$ and $Y$ be Gaussian with variances $\sigma_U$, $\sigma_W$, $\sigma_Z$, $\sigma_X$ and $\sigma_Y$, respectively, to be able to obtain closed-form expressions for the quantities of interest (see Appendix C).

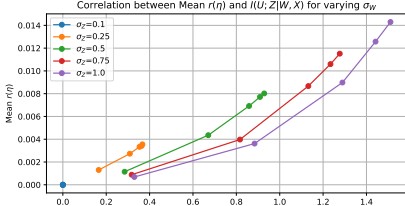

Figure 2 shows the results for the relative approximation error $r(\eta) = \mathbb{E}_{Z \sim p(Z|x)}[\eta/|\mathbb{E}[Y|x,Z]|]$ vs. $I(U;Z|W,x)$ both averaged over $X$, for $\sigma_U = 10$, $\sigma_Z, \sigma_W = \{0.1, 0.25, 0.5, 0.75, 1\}$, and $\sigma_X = 0.1$, from which we see that $\eta$ increases with $I(U;Z|W,x)$ and $\sigma_W$, consistent with Theorem 3. We provide more details and results varying $\sigma_X$ in Appendix D. Interestingly, when $\sigma_X = \sigma_Z$ and the SEM coefficients are 1, both $\eta = 0$ and $I(U;Z|W,x) = 0$, which is a special case formalized in Lemma 1 in Appendix C.

Figure 2: Relative error ($r(\eta)$) vs. mutual information ($I(U;Z|W,x)$) both averaged over $X$. Each line represents a value of $\sigma_Z$ for increasing values of $\sigma_W = \{0.1, 0.25, 0.5, 0.75, 1\}$, $\sigma_X = 0.1$, which are consistent with $I(U;Z|W,x)$.

## 5 Generative Model for the Bridge Function

Concerning the assumed form of $b(W,x)$ in Theorem 3, while $p(U|W,x,z)$ was replaced with $p(U|W,x)$, $\mathbb{E}(Y|x,W,U)$ was retained from $b_0(W,x,z)$. The latter was conducive for analysis. However, it is possible that a better bridge $b(W,x)$ may be learned without retaining $\mathbb{E}(Y|x,W,U)$, yielding a smaller expected difference between $\mathbb{E}(Y|x,z)$ and $\mathbb{E}[b(W,x)|x,z]$. We therefore can replace 2 with the following general form which we use in our model:

$$b(W,x) = \int dU \ g(x,W,U)p(U|W,x), \tag{10}$$

which allows $b(W,x)$ to not be restricted to $g(x,W,U) = \mathbb{E}[Y|x,W,U]$, while still being able to produce the desired $\mathbb{E}[Y|x,z]$. Both $g(x,W,U)$ and $p(U|W,x)$ are jointly learned when solving (1)). Note that this setup effectively suggests a generative latent-variable model with encoder $p(U|W,x)$ and a decoder for $(X,Z)$, which we discuss in detail in Section 6. In practice we do not claim the modeled $p(U|W,x)$ represents truth, as final predictions are based on $b(W,x)$.

We seek to model the form of the bridge as in (10). From that perspective, a generative model can be constructed to draw the samples of $U$ needed for implementing (1). In this context, and using (10)), we consider $u_j \sim p(U|w_j,x)$ with $w_j \sim p(W|x,z)$. One may consider *learning* a function $h(W,x,\epsilon)$, such that $u_j = h(w_j,x,\epsilon_j)$, where $w_j \sim p(W|x,z)$ and (for example) $\epsilon_j \sim \mathcal{N}(0,I)$, where $I$ is the identity matrix and hence $\epsilon$ is drawn from isotropic Gaussian noise of the chosen dimension. Such a generative model $h(W,x,\epsilon)$ has been widely considered, *e.g.*, in generative adversarial networks (GANs) and its generalizations [18, 34, 3, 12, 32]. In practice, using Proposition 2, we may model

$$\mathbb{E}(Y|x,z) = \mathbb{E}_{W|x,z}\Big[\underbrace{\mathbb{E}_{p(\epsilon)}[g_Y(x,W,h(W,x,\epsilon))]}_{b(W,x)}\Big], \tag{11}$$

which is motivated by modeling $b(W,x)$ in such a way that the expectation-matching property of (1) holds, but explicitly defining $b(W,x)$ as a functional expectation over $p(U|W,x)$ as in (11), modeled here by samples drawn through $h(W,x,\epsilon)$, with a general integrand $g_Y(x,W,U)$. The benefit of this model is most prominent when it is coupled with an autoencoder for $(X,Z)$, in which $h(W,x,\epsilon)$ is shared, thus enhancing statistical strength. This strategy is discussed next.

## 6 Learning Setup

Assume that we have access to a set of data $\mathcal{D}_1 = \{(x_i,z_i,w_i)\}_{i=1,M}$ from which a generative model can be learned to draw samples from $p(W|x,z)$. The details of this model depend on the data characteristics, and more details are presented when discussing the experiments in Section 7.

Note that we need not model $p(W|x, z)$, rather in general, we seek to model the capacity to generate samples from this conditional distribution, *e.g.*, using a conditional GAN [32].

We also assume access to data $\mathcal{D}_2 = \{(x_i, z_i, y_i)\}_{i=1,N}$, which may or may not be explicitly connected to $\mathcal{D}_1$. We use $\mathcal{D}_2$ within the Fredholm integral in (1) with which we will solve for $b(W, x)$, with conditional expectations wrt $p(W|x, z)$ performed by drawing samples from the generative model developed with $\mathcal{D}_1$. We note that $\mathcal{D}_1$ is not necessary; however, it is specified to weaken the requirement of having a common dataset for which all observed variables $(W, Z, X, Y)$ are available. This distinction is especially useful in situations where the dataset for which the outcome is available $\mathcal{D}_2$ is not the same or is (much) smaller than $\mathcal{D}_1$.

**Bridge for the Outcome** Let $g_{\theta_Y}(x, W, h(W, x, \epsilon))$ represent a model for $\mathbb{E}(Y|x, W, U)$, where we model samples of the unobserved confounder as $U = h_{\theta_U}(W, x, \epsilon)$, $\theta_Y$ and $\theta_U$ are the model parameters with subscripts $Y$ and $U$ highlighting that the model is connected to expected outcomes and the unobserved confounder, respectively. To solve (1), we seek to minimize the following loss.

$$\mathcal{L}_{\theta_Y} = \sum_{i=1}^{N} \left( y_i - \mathbb{E}_{p(W|x_i, z_i)} \mathbb{E}_{p(\epsilon)} [g_{\theta_Y}(x_i, W, h_{\theta_U}(W, x_i, \epsilon))] \right)^2. \tag{12}$$

If only the outcome $Y$ is modeled, then in practice we replace $\mathbb{E}_{p(\epsilon)}[g_{\theta_Y}(x, W, h_{\theta_U}(W, x, \epsilon))]$ with $b(W, x)$, *e.g.*, a neural network with inputs $(W, x)$, which is understood to have parameters $\theta_Y$.

The objective implied by $\mathcal{L}_{\theta_Y}$ involves two steps: (*i*) develop a generative model to draw samples from $p(W|x, z)$ using $\mathcal{D}_1$, and (*ii*) use this model within the minimization of $\mathcal{L}_{\theta_Y}$ for $\theta_Y$ to approximate the expectation wrt $W$. However, note that these two steps are followed in sequence, which should be distinguished from the *iterative and alternating* two-step approach developed in [52].

Within the context of our discussion after (10), through $w_j \sim p(W|x, z)$ and $\epsilon_j \sim p(\epsilon)$, we seek to simulate samples from $p(U|x, z)$ as $u_j = h_{\theta_U}(w_j, x, \epsilon_j)$. The unobserved $U$ is assumed to affect both the treatment $X$ and $Z$ as illustrated in Figure 1. When the number of samples $N$ is relatively small, there may be an opportunity to improve statistical strength by also modeling $\{(x_i, z_i)\}_{i=1,N}$, both of which are also functions of $U$ (not only $Y$).

**Autoencoder for the Treatment and its Control** Analogous to the aforementioned model for $Y$, we can model $\mathbb{E}(X|U, z) = g_{\theta_X}(U, z)$ and $\mathbb{E}(Z|U) = g_{\theta_Z}(U)$. In the context of an autoencoder, we assume that $u_j = h_{\theta_U}(w_j, x_i, \epsilon_j)$ with $w_j \sim p(W|x_i, z_i)$ constitutes a means of encoding observations $(x_i, z_i)$ into a latent feature space represented by conditional samples $u_j|(x_i, z_i)$. Connected to the approximation considered in Proposition 2, we assume that the expectations of $g_{\theta_X}(U, z)$ and $g_{\theta_Z}(U)$ wrt $p(U|W, x, z)$ may be replaced by expectations wrt $p(U|W, x)$ with $W \sim p(W|x, z)$. This implies the same conditions on $p(U|W, x, z)$, *e.g.*, that the expectation of $p(U|W, x, z)$ depends only on $(W, x)$, and that $g_{\theta_X}(U, z)$ and $g_{\theta_Z}(U)$ have the same class of dependence on $U$ as $g_{\theta_Y}(x, W, U)$, *e.g.*, they could be linear in $U$. However, although these functions may be linear in $U$, $U$ itself may be nonlinearly related to $W$ (see Corollary 1). From this angle, we consider the additional losses:

$$\mathcal{L}_{\theta_X} = \sum_{i=1}^{N} \left( x_i - \mathbb{E}_{p(W|x_i, z_i)} \mathbb{E}_{p(\epsilon)} [g_{\theta_X}(h_{\theta_U}(W, x_i, \epsilon), z_i)] \right)^2$$
$$\mathcal{L}_{\theta_Z} = \sum_{i=1}^{N} \left( z_i - \mathbb{E}_{p(W|x_i, z_i)} \mathbb{E}_{p(\epsilon)} [g_{\theta_Z}(h_{\theta_U}(W, x_i, \epsilon))] \right)^2, \tag{13}$$

where we emphasize that the function $h_{\theta_U}(W, x, \epsilon)$ parameterized by $\theta_U$ is *shared* between the models of $(Y, X, Z)$, ideally improving the quality of the learned $h_{\theta_U}(W, x, \epsilon)$. Note that when producing causal estimates, we only need the learned $g_{\theta_Y}(x, W, h(W, x, \epsilon))$, *i.e.*,

$$\mathbb{E}(Y|do(X = x)) = \mathbb{E}_{p(W)} \mathbb{E}_{p(\epsilon)} [g_{\theta_Y}[x, W, h_{\theta_U}(W, x, \epsilon)]]. \tag{14}$$

However, by jointly seeking to minimize $\mathcal{L}_{\theta_Y} + \mathcal{L}_{\theta_X} + \mathcal{L}_{\theta_Z}$, it is hoped that the quality of the model for $h_{\theta_U}(W, x, \epsilon)$ will be improved, therefore also improving the causal-effect estimates relative to the outcomes. In practice, weighting can be used in the sum of losses, to reflect their relative scale which depends on $(X, Z, Y)$, as discussed in Section 7. Moreover, in terms of implementation, we still first build a generative model for $p(W|x, z)$ using $\mathcal{D}_1$ and then optimize the parameters of the shared encoder $\theta_U$, bridge $\theta_Y$ and autoencoding components $\{\theta_X, \theta_Z\}$ using $\mathcal{D}_2$.

**Connection to the Causal Effects VAE (CEVAE)** The composite model, employing the *cumulative* loss function $\mathcal{L}_{\theta_Y} + \mathcal{L}_{\theta_X} + \mathcal{L}_{\theta_Z}$, shares characteristics and motivation with the CEVAE developed in [28]. We note that recent work by [38, 52] has highlighted limitations of the CEVAE. Some of the challenges with existing CEVAE research are related to the difficulty of modeling posteriors like $p(U|x, z)$.

The proposed framework differs from the CEVAE in several key ways. Within the context of (10), $W$ is assumed to be a strong proxy for $U$, and therefore the model to draw samples from $p(W|x,z)$, based on *observed* $\{(x_i, z_i, w_i)\}_{i=1,M}$ provides strong information about $p(U|x,z)$ that was not available to the CEVAE. Moreover, within our autoencoder, we only model the latent confounder with $(x_i, z_i)$ and *not* the outcomes $y_i$ as in the CEVAE. The expected conditional outcome $\mathbb{E}(Y|x,z)$ is modeled via the causal bridge function, for which we have theoretical support (Theorem 3).

Finally, note that within the CEVAE one must set a prior $p(U)$ with which the Kullback-Leibler (KL) divergence is computed relative to $p(U|x_i, z_i)$. In practice, it can be difficult to set such a prior; thus we avoid this complication by simply designing an autoencoder, instead of *a variational* autoencoder. Rather than employing a KL term for a regularization of the posterior, we use $\mathcal{L}_{\theta_Y}$ and our model to draw samples from $p(W|x_i, z_i)$, both of which provide regularization on the inferred posterior.

The extension of our approach to a CEVAE-type setup is relatively straightforward, using our definition of $p(U|x,z)$ within the CEVAE. Importantly, in such a setting, the CEVAE models $(X, Z)$, and $Y$ is handled via the Fredholm equation for the bridge. The main difference between a CEVAE version of our approach is the inclusion of a prior $p(U)$ and the inclusion of a KL term between $p(U)$ and $p(U|x,z)$. Nevertheless, we did implement such a modified CEVAE formulation, in addition to the simpler autoencoder setup discussed above. We found that the KL term added significant difficulty and undermined the reliability of CEVAE-based predictions. We do not show results for an CEVAE-like version of our model (with KL term) in Section 7, because they were numerically unstable and sensitive to the way $p(U)$ was set (like shown in [52]). In contrast, we found that our approach trained well and yielded reliable results. However, in Section 7 we do discuss and compare to results from the original CEVAE model [28].

**Bridge for Survival (Time to Event) Outcomes** So far we have considered continuous outcomes $Y$ with standard squared error loss $\mathcal{L}_{\theta_Y}$. We now consider survival outcomes, which are of special interest in a wide range of scenarios where causal inference is used in practice, constituting a novel application of the causal-bridge framework. Specifically, we consider outcomes of the form $(Y, E)$, where $Y$ is the observed time $Y = \min(T, C)$, $T$ is the time at which the event of interest occurs, $C$ is the follow-up time, and $E$ is the observed-event indicator. If for a given sample, $y = t < c$ it is said that the event of interest is observed and $e = 1$, otherwise, $y = c < t$, $e = 0$ and the event is right-censored. Here we assume that censoring is not informative, *i.e.*, $T \perp\!\!\!\perp C|X$. Extensions to other forms of censoring are possible within our framework, but left as future work. Unlike for continuous outcomes, we are not interested in modeling the (expected) value of $Y$ through $\mathbb{E}[Y|do(X = x)]$. Instead, we are interested in $\mathbb{E}[\lambda|do(X = x)]$, *i.e.*, the risk function defined as the contribution of observed covariates on a baseline hazard function, *i.e.*, $\lambda(Y|W, x) = \lambda_0(t)\exp(b(W, x))$, where $\lambda_0(t)$ is the baseline hazard and $\exp(b(W, x))$ is the risk function, conveniently written in terms of a bridge function. The casual estimate of interest for binary treatments, $X = \{0, 1\}$ is the hazards ratio (HR) defined as

$$\text{HR} = \frac{\lambda(Y|W, X = 1)}{\lambda(Y|W, X = 0)} = \frac{\exp(b(W, X = 1))}{\exp(b(W, X = 0))}. \tag{15}$$

Optimizing (1) wrt the hazards function is achieved by maximizing the partial likelihood of the model similar to the Cox proportional hazard model [13] using

$$\mathcal{L}_{\theta_Y} = \sum_{i:e_i=1} \rho_i - \log\Big(\sum_{j:y_j>y_i} \exp(\rho_i)\Big), \; \rho_i = \mathbb{E}_{p(W|x_i, z_i)}\mathbb{E}_{p(\epsilon)}[g_{\theta_Y}(x_i, W, h_{\theta_U}(W, x_i, \epsilon))]. \tag{16}$$

where we do not need to account for the baseline hazards $\lambda_0(t)$ because it does not depend on the treatment $X$ or the outcome control $W$. Note that the autoencoder version of this model is consistent with the above definition, except that the loss for $X$ is changed to cross entropy.

Recently, [54] proposed an approach that also modeled the hazard function using the bridge function. However, they do not make the proportional hazard assumption like we do. Rather, they impose a rigid form for the bridge function (Eq. (31) in [54]). That work considered experiments on real data (SUPPORT), as we do. However, in [54] the ground truth for the estimate of the causal effect is not available, while in our experiments in Section 7 we compare to an RCT.

# 7 Experiments

All models were developed using PyTorch, and each experiment can be executed in a few minutes on a Tesla V100 PCIe 16 GB GPU. The source code used here can be found at `https://github.com/ruolinmeng/CausalBridgeAutoEncoder`.

## 7.1 Synthetic data

We first demonstrate the performance of the proposed method on two synthetic datasets introduced in [52], and we perform the same experiments as considered there.

**Data** The first experiment considers the *Demand* data introduced by [19]. Details of the data generation process are found in Appendix E.1. The second experiment considers the *dSprite* data introduced by [29]. Details of the data generation process can be found in Appendix E.2. For both datasets, we consider two sample size settings, 1000 and 5000, for a single dataset $\{(x, z, w, y)\}$, consistent with the setup in [52].

**Metrics** We estimate the average causal effect using the bridge function with (14) and compare it to the ground-truth causal effect in terms of the mean squared error (MSE). Note that this is only possible with synthetic data since the datasets used to train the model only have access to specific (*factual*) combinations of $\{(x, z, y)\}$, *i.e.*, we do not also have values of $\{(z, y)\}$ for which the treatment $x$ takes values different (*counterfactual*) than those observed. The test points are evenly distributed over the treatment variable $X$ when calculating the out-of-sample MSE. Additional details are provided in Appendix E. This setup replicates the experiments introduced by [52].

**Models considered** We compare our method to the deep feature proxy variable (DFPV) method of [52], which is considered the prior state of the art. In the results below, we consider the following model configurations: $i$) The original DFPV method [52]. $ii$) The same underlying DFPV model (and hyperparameters), but replace iterative learning [52] by first estimating $p(W|x, z)$, from which we sample when learning the bridge function. This setup allows examination of the benefits of learning to sample from $p(W|x, z)$, with no change to the form of $b(W, x)$ from DFPV. $iii$) The causal bridge (CB) model learned with (12) and using the $p(W|x, z)$ sampling model. $iv$) The combined bridge and autoencoder (CB + AE) model with (12) and (13), *i.e.*, $\mathcal{L}_{\theta_Y} + \mathcal{L}_{\theta_X} + \mathcal{L}_{\theta_Z}$. Moreover, we also consider CEVAE [28], neural maximum moment restriction (NMMR) with U- or V-statistic optimization so NMMR-U and NMMR-V, respectively [24], and Kernel Alternative Proxy (KAP) [8]. Concerning the new methods discussed in Section 6 (the last two methods above), our model architectures are relatively simple and not considerably more complex than those used by DFPV [52]. This is done to demonstrate that the performance of our model variants is not attributed to overly complicated architectures. The details of the neural networks and hyperparameter tuning for all models are provided in Appendix F. Concerning the last two models discussed above: CB learned based on modeling $Y$ alone and CB + AE is based on the combined loss for modeling $(Y, X, Z)$, the *exact same models and hyperparameters* are used for $g_{\theta_Y}(x, W, h_{\theta_U}(W, x, \epsilon))$ (this allows consideration of the impact of the autoencoder, without anything else changed). We also note that model selection for all approaches is based only on the loss for $Y$ in (12).

In all places where we sample from $p(W|x, z)$, 100 samples are drawn. For the model $h_{\theta_U}(W, x, \epsilon)$, for each draw of $W$, 5 unique $\epsilon$ are drawn from $p(\epsilon)$. These values were selected based on empirical validation to ensure stable learning and accurate estimation.

**Results** Figure 3 shows that for experiments with synthetic data, Demand and dSprites, using our generator for $p(W|x, z)$ significantly improves performance relative to DFPV [52], which is likely due to the advantages of sampling (we draw 100 samples from $p(W|x, z)$ in these experiments). Consequently, we have the flexibility to generate many samples of $W$ for each combination of $\{(X = x, Z = z)\}$, enabling better learning and estimation of causal effects. Moreover, utilizing $h_{\theta_U}(W, x, \epsilon)$ in the generalized bridge model $g_{\theta_Y}(x, W, h_{\theta_U}(W, x, \epsilon))$, and thus moving beyond the DFPV representation of the bridge, further improves performance. The gains of this approach using (12) are most noticeable on the Demand dataset. Incorporating the autoencoder for $(X, Z)$ using *both* (12) and (13) aids with the learning of the shared $h_{\theta_U}(W, x, \epsilon)$, particularly when the sample size is small, which is of particular interest in real-world scenarios. Finally, when comparing the proposed CB and CB + AE with NMMR-U, NMMR-V and KAP, we see that our approaches are significantly better, especially on the dSprites dataset. Results for KAP on Demand and CEVAE on both datasets

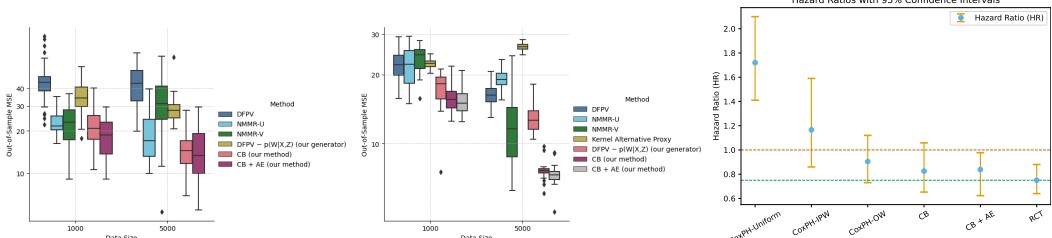

Figure 3: Out-of-sample MSE results for (Left) **Demand** and (Middle) **dSprite** data. (Right) Hazard-ratio (HR) results with 95%CIs for **Framingham** data. The different methods are listed along the x axis, including results from the RCT, to which CB + AE agrees best. The red and green dashed lines correspond to the null HR= 1 and the reference (mean RCT estimate), respectively.

are not shown because they are out of range, however, they are presented in Appendix Table 1 along with all results in Figure 3 for completeness.

In Appendices I and J we present two ablation studies. In the former, we explore the dimensionality of $U$ and noise $\epsilon$ for the shared autoencoder $h_{\theta_U}(W, x, \epsilon)$, by showing in Figure 12 that our model is insensitive to them. In Appendix J we explore the generalization in (10), but instead of using $h_{\theta_U}(W, x, \epsilon)$ to sample from $p(U|W, x)$, we do it directly via MCMC (we can do this when we assume the *unrealistic* case of knowing the underlying model – we do this as a test of the utility of the model $h_{\theta_U}(W, x, \epsilon)$). We show in Figure 13 that learning $g(x, W, U)$ with samples from $p(U|W, x)$ produces slightly more accurate causal estimates compared to using $h_{\theta_U}(W, x, \epsilon)$, which is a good indication that the latter is a good approximation for $p(U|W, x)$ for the specific purpose of estimating causal effects.

## 7.2   Real-World data

**Data** Framingham is an observational longitudinal study designed to learn about the incidence and prevalence of cardiovascular disease (CVD) and its risk factors [5]. The data used here are the *Offspring cohort*, which consists of 3435 subjects split into 2404, 516, and 515 training, validation, and test samples, respectively. For this experiment, we are interested in estimating the average causal effect of taking *statins* (a cholesterol-lowing medication), *i.e.*, the treatment $X = \{0, 1\}$, on the timing $Y$ of future CVD events. These data are in the public domain, and therefore these experiments can be replicated.

**Constructing proxies for Framingham** The 32 covariates available for this dataset are not split in terms of treatment and outcome controls, $Z$ and $W$, respectively. Consequently, we adopt the proxy bucketing strategy of [47], which divides covariates into $Z$ and $W$ according to their association with treatments and outcomes using effect sizes estimated from linear models on the training data. Details are provided in Appendix E.3.

**Models considered** We compare the proposed framework with three strong baselines: variants of balancing weighting schemes for the Cox proportional hazards (CoxPH) model for survival analysis [42]. Balancing weights are obtained from a linear logistic regression model built to estimate propensity scores $s = P(X = 1|W = w, Z = z)$. Then, these weights are used to fit a CoxPH model where the input is *only* the treatment and the objective uses the balancing weights. We consider three weighting schemes: (CoxPH-Uniform) where all weights are 1; (CoxPH-IPW) using inverse probability weighting, $\omega = x/s + (1 - x)/(1 - s)$ [10]; (CoxPH-OW) using overlapping weighting, $\omega = x(1 - s) + (1 - x)s$ [27]. We also consider the two proposed variants of the causal bridge, *i.e.*, CB and CB + AE. Additional details about the CoxPH model and overlapping weights are provided in Appendix G. The sampler $p(W|x, z)$, the bridge and autoencoder architectures, and model selection details are provided in Appendix F.

**Randomized control trial** We also report the HR estimated from a separate randomized control trial (RCT) conducted specifically to assess the effects of statins on CVD outcomes [56]. These RCT results provide a powerful reference against which causal estimates can be compared.

**Metrics** We estimate the causal effect of the treatment using the *hazard ratio* (HR). For the CoxPH model, the HR is obtained simply as $\exp(\beta)$ where $\beta$ is the coefficient for the treatment obtained

from fitting the weighted model. For the proposed bridge model we use the strategy described in Section 6. HR values are interpreted as positive (HR$< 1$), negative (HR$> 1$) or neutral (HR$\approx 1$) in relation to the effect of the treatment on the outcome. Confidence intervals (95% CIs) for the HR are obtained using asymptotic estimates for CoxPH model [13] and empirical quantile estimates (from multiple model runs and samples of $W$) for the proposed model. Additional details are provided in Appendix G. For completeness, in Figure 15 in Appendix K we also report the concordance index (C-Index) on the test set, which is a widely used metric to assess the predictive power of survival models. Note that we cannot obtain a C-Index for CoxPH variants because these are built only using treatment ($X$) and outcome ($Y$), thus unable to produce predictions.

**Results** Figure 3 shows HR estimates with 95%CIs indicating that the CB approaches outperform the CoxPH variants. We note that CoxPH-Uniform and CoxPH-IPW both result in HR$> 1$ indicating that the treatment increases the risk of CVD events, which occurs because subjects at high risk of CVD are the ones treated with statins (see Figure 14 in Appendix K), thus effectively confounding its effect on the outcome when using observational data. The other three estimates (CoxPH-OW, CB and CB + AE) result in HR$< 1$, however, notably, CB + AR results in tighter estimates with a 95% CI away from HR$= 1$. Importantly, the causal-bridge results are consistent with expectations from the RCT. For completeness, Figure 15 in Appendix K show results for CB and CB + AE as boxplots.

## 8 Conclusions

We have introduced a framework for causal inference with control variables by re-examining core assumptions connected to the causal bridge. This yielded a new bound on the error of the causal bridge, providing insights into the estimation of causal effects with unobserved confounders. Our approach employed a conditional generative model for $W$, and motivated a new framing of the model by the inclusion of an autoencoder. We also extended the causal-bridge framework to survival analysis, broadening its applicability to a wider range of real-world problems. Empirical validation on both synthetic and real-world datasets confirms the superior performance of our proposed approach compared to state-of-the-art methods. As an interesting direction for future work, we recognize that it is possible to derive a similar result presented here for the causal (outcome) bridge but for the treatment confounding bridge using ideas from [14], although noting that such a treatment bridge is most useful for binary treatments (interventions), while most of our experiments considered continuous (real-valued) treatments. Moreover, we can also extend the proposed methodology to scenarios where post-treatment variables are available like in [49].

**Limitations** This work has several limitations, $i$) analogous to other causal inference frameworks, verifying the assumptions is difficult; $ii$) defining or splitting covariates into proxies, $W$ and $Z$, may be challenging in practical scenarios; $iii$) we acknowledge that the proportional hazard assumption used for the time-to-event model can be seen as a limitation; however, one can alternatively consider a specific formulation for it like in [54], however, such an approach has also issues related to model misspecification that are difficult to address in practice; and $iv$) we understand that having additional real-world datasets will make the experimental results stronger; however, it is difficult to find real-world datasets for which the ground-truth causal effects are known, such as it is for the case shown in our experiments using the Framingham dataset.

## Acknowledgments

This work was supported by ONR grant number 313000130.

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

## Summary of Appendix Content

This appendix provides an extensive set of details associated with the material in the paper. To aid the reader in navigating and using this Appendix, we summarize what is provided and in what section (with a hyperlink to it).

## A  Broader Impact Statement

This research involves the estimation of causal treatment effects, using proxy variables for the real-world survival dataset. Even though the proposed approach is not a replacement for direct measurements of drug effects, more accurate causal inference from real-world data can improve clinical decision making by supplementing it. Although this work has the potential to enhance drug impact assessment, careful application is essential to avoid unexpected outcomes. From a theoretical perspective, this study seeks to advance the generalization of causal inference using proxy data, which is increasingly vital for machine learning models, particularly in high-stakes decision-making domains such as healthcare, finance, and policy. Thus, it includes the societal consequences, ranging from ethical to environmental considerations linked to the field of machine learning.

## B  Proofs (Theorem 3, Corollary 1, and Concentration Inequality)

### B.1  Proof of Theorem 3

We first consider the case $Z = z$, and then we will average over $z$. Fix $x$ and $z$. For each $W$, define

$$\delta(W) := \mathbb{E}_{U \sim p(U|W,x)}[f(U)] - \mathbb{E}_{U \sim p(U|W,x,z)}[f(U)],$$

where $f(U) = \mathbb{E}[Y|x, W, U]$. Since $f$ is $C$-Lipschitz and $U$ is supported on a set of radius $R$, it follows from Pinsker's inequality and the standard variational form of KL divergence (see [50, 37]) that:

$$|\delta(W)| \leq CR \cdot \sqrt{2D_{\mathrm{KL}}(p(U|W,x,z) \,\|\, p(U|W,x))}.$$

This inequality is related to a result in the proof of Lemma 1 in [50], using here that a $C$-Lipschitz function $f(U)$, with $U$ supported on a ball of radius $R$, is $CR$-subGaussian [48].

Now take the expectation over $W \sim p(W|x,z)$:

$$\left| \mathbb{E}_{W \sim p(W|x,z)}[b(W,x) - b_0(W,x,z)] \right| \leq \mathbb{E}_{W \sim p(W|x,z)}|\delta(W)|.$$

Apply Jensen's inequality to the concave square root function, and the series of inequalities continues:

$$\mathbb{E}_{W \sim p(W|x,z)}|\delta(W)| \leq CR \cdot \mathbb{E}_{W \sim p(W|x,z)} \left[ \sqrt{2D_{\mathrm{KL}}(p(U|W,x,z) \,\|\, p(U|W,x))} \right]$$

$$\leq CR \cdot \sqrt{2 \, \mathbb{E}_{W \sim p(W|x,z)} \left[ D_{\mathrm{KL}}(p(U|W,x,z) \,\|\, p(U|W,x)) \right]}.$$

From above, we have a result for fixed $z$. We now take expectation over $Z \sim p(Z|x)$ on both sides:

$$\mathbb{E}_{p(Z|x)} \left[ \left| \mathbb{E}_{p(W|x,Z)}[b(W,x) - b_0(W,x,Z)] \right| \right]$$

$$\leq CR \cdot \mathbb{E}_{p(Z|x)} \left[ \sqrt{2 \, \mathbb{E}_{p(W|x,Z)} \left[ D_{\mathrm{KL}}(p(U|W,x,Z) \,\|\, p(U|W,x)) \right]} \right].$$

Now apply Jensen's inequality (the square root is concave) to move the outer expectation inside the square root, the subsequent inequality follows:

$$\leq CR \cdot \sqrt{2\, \mathbb{E}_{p(W,Z|x)}\left[D_{\mathrm{KL}}(p(U|W,x,Z) \,\|\, p(U|W,x))\right]}.$$

The right-hand side becomes:

$$CR \cdot \sqrt{2\, \mathbb{E}_{p(W,Z|x)}\left[D_{\mathrm{KL}}(p(U|W,x,Z) \,\|\, p(U|W,x))\right]} = CR \cdot \sqrt{2I(U;Z|W,x)}.$$

which is the statement of Theorem 3, recalling that $\mathbb{E}(Y|x,z) = \mathbb{E}_{W \sim p(W|x,z)}[b_0(x,W,U)]$.

Providing more detail, the final step states:

$$\mathbb{E}_{p(W,Z|x)}[D_{KL}(p(U|W,x,Z)\|p(U|W,x))] = I(U;Z|W,x)$$

To show that, consider that the conditional mutual information $I(U;Z|W,x)$ is defined as:

$$I(U;Z|W,x) = \int\int\int p(u,z,w|x)\log\left[\frac{p(u,z|w,x)}{p(u|w,x)p(z|w,x)}\right]\,du\,dz\,dw \qquad (17)$$

$$= \int\int\int p(u,z,w|x)\log\left[\frac{p(u|w,z,x)}{p(u|w,x)}\right]\,du\,dz\,dw \qquad (18)$$

This can be rewritten as:

$$I(U;Z|W,x) = \int\int p(w,z|x)\left[\int p(u|w,z,x)\log\left[\frac{p(u|w,z,x)}{p(u|w,x)}\right]\,du\right]\,dw\,dz$$

The inner integral is precisely the KL divergence $D_{KL}(p(U|W,Z,x)\|p(U|W,x))$, so:

$$I(U;Z|W,x) = \int\int p(W,Z|x)\, D_{KL}(p(U|W,Z,x)\|p(U|W,x))\,dW\,dZ \qquad (19)$$

$$= \mathbb{E}_{p(W,Z|x)}[D_{KL}(p(U|W,Z,x)\|p(U|W,x))] \qquad (20)$$

## B.2 Complete Statement and Proof of Corollary 1

**Corollary 1:** *Suppose that the random variables $(U, Z, X, W)$ satisfy the following conditions:*

(i) *$W = \Psi(U) + \epsilon$, where $\Psi : \mathbb{R}^{d_U} \to \mathbb{R}^{d_W}$ is an invertible, continuously differentiable ($C^1$) function,*

(ii) *$\epsilon$ is independent of $(U, Z, X)$, with zero mean and covariance matrix $\sigma_\epsilon^2 I$,*

(iii) *The latent confounder $U$ has finite differential entropy $h(U) < \infty$,*

(iv) *The conditional distribution $p(Z|U,x)$ is Lipschitz in $U$; specifically, there exists $L_{Z,U,x} > 0$ such that*

$$|\log p(Z|u_1,x) - \log p(Z|u_2,x)| \leq L_{Z,U,x}\|u_1 - u_2\|, \quad \forall u_1, u_2.$$

(v) *$Z \perp\!\!\!\perp W \mid (U,x)$*

(vi) *$Q_{\min} := \min_{(U,Z)} p(U \mid W,x) \cdot p(Z \mid W,x) > 0$.*

*Then, there exists a constant $C_0 > 0$ depending on the Lipschitz constants of $\Psi$ and $p(Z|U)$ such that, for sufficiently small $\sigma_\epsilon$,*

$$I(U;Z|W,x) \leq C_0\sigma_\epsilon^2.$$

*In particular,*

$$\lim_{\sigma_\epsilon^2 \to 0} I(U;Z|W,x) = 0.$$

**Proof:**
Since $W = \Psi(U) + \epsilon$ and $\Psi$ is invertible and $C^1$, we can locally linearize $\Psi$ around any $U$. For small noise $\epsilon$, we have the Taylor expansion:

$$W = \Psi(U) + \epsilon \quad \Rightarrow \quad U \approx \Psi^{-1}(W - \epsilon).$$

Expanding $\Psi^{-1}(W - \epsilon)$ around $W$ gives:

$$\Psi^{-1}(W - \epsilon) = \Psi^{-1}(W) - \nabla\Psi^{-1}(W)\epsilon + o(\|\epsilon\|),$$

where $\nabla\Psi^{-1}(W)$ is the Jacobian of $\Psi^{-1}$ at $W$.

Thus, conditioned on $W$, the distribution of $U$ is centered around $\Psi^{-1}(W)$, with fluctuations proportional to $\epsilon$. To be more specific, as $\sigma_\epsilon \to 0$, we consider following approximation

$$p(U \mid W, x) \xrightarrow{w} \delta_{\mu^*(W)}(U) \tag{21}$$

where $\xrightarrow{w}$ refers to weak convergence, $\mu^*(W) := \Psi^{-1}(W)$ and $\delta_{\mu^*(W)}$ is Dirac function centered at $\mu^*(W)$. Moreover, the conditional covariance of $p(U \mid W, x)$ can be approximated as

$$\mathrm{Var}(U|W, x) \approx \nabla\Psi^{-1}(W)\,\mathrm{Var}(\epsilon)\left(\nabla\Psi^{-1}(W)\right)^\top = O(\sigma_\epsilon^2), \tag{22}$$

because $\mathrm{Var}(\epsilon) = \sigma_\epsilon^2 I$, and $\nabla\Psi^{-1}(W)$ is bounded by the invertibility and smoothness of $\Psi$.

With above approximation, we have

$$\begin{aligned}
p(Z \mid W, x) &= \int p(Z \mid U, x)p(U \mid W, x)dU &&\text{(Using } Z \perp\!\!\!\perp W \mid (U, x)) \\
&\approx \int p(Z \mid U, x)\delta_{\mu^*(W)}(U)dU &&\text{(Using (21))} \\
&= p(Z \mid \mu^*(W), x). \tag{23}
\end{aligned}$$

Then, by Lipschitz assumption on $\log p(Z \mid U)$, we can derive

$$\begin{aligned}
|p(Z \mid W, x) - p(Z \mid \mu^*(W), x)| &= |\int [p(Z \mid U, x) - p(Z \mid \mu^*(W), x)] \cdot p(U \mid W, x)dU| \\
&&&\text{(Using } Z \perp\!\!\!\perp W \mid (U, x)) \\
&\leq \int |p(Z \mid U, x) - p(Z \mid \mu^*(W), x)| \cdot p(U \mid W, x)dU \\
&\leq L_{Z,U,x} \int |U - \mu^*(W)|p(U \mid W, x)dU \\
&= L_{Z,U,x} \cdot \mathbb{E}_{U|W,x}[|U - \mu^*(W)|]. \tag{24}
\end{aligned}$$

Notice that

$$\begin{aligned}
\mathbb{E}_{U|W,x}[|U - \mu^*(W)|] &= \mathbb{E}_{U|W,x}[\sqrt{|U - \mu^*(W)|^2}] \\
&\leq \sqrt{\mathbb{E}_{U|W,x}[|U - \mu^*(W)|^2]} &&\text{(Using Jensen's inequality)} \\
&= \sqrt{\mathrm{Var}(U|W, x)} \approx O(\sigma_\epsilon). &&\text{(Using (22))} \tag{25}
\end{aligned}$$

Hence, we have

$$|p(Z \mid W, x) - p(Z \mid \mu^*(W), x)| \leq L_{Z,U,x} \cdot O(\sigma_\epsilon). \tag{26}$$

Now, observe that the $p(U, Z|W, x)dUdZ$ is close to the factorized distribution $p(U|W, x)p(Z|W, x)dUdZ$, because

$$\begin{aligned}
p(U, Z \mid W, x)dUdZ &= p(U \mid W, x)p(Z \mid U, x)dUdZ &&\text{(Using } Z \perp\!\!\!\perp W \mid (U, x)) \\
&\approx \delta_{\mu^*(W)}(U)p(Z \mid U, x)dUdZ &&\text{(Using (21))} \\
&= \delta_{\mu^*(W)}(U)p(Z \mid \mu^*(W), x)dUdZ \\
&\approx p(U \mid W, x)p(Z \mid W, x)dUdZ. &&\text{(Using (23))} \tag{27}
\end{aligned}$$

We consider $P = p(U, Z \mid W, x)$, $Q = p(U \mid W, x) \cdot p(Z \mid W, x)$. The total variation distance would be

$$\begin{aligned}
\mathrm{TV}(P, Q) &= \frac{1}{2} \int |p(U, Z \mid W, x) - p(U \mid W, x) \cdot p(Z \mid W, x)| dU\, dZ \\
&= \frac{1}{2} \int |[p(Z \mid U, x) - p(Z \mid W, x)] \cdot p(U \mid W, x)| dU\, dZ \\
&\approx \frac{1}{2} \int |[p(Z \mid U, x) - p(Z \mid W, x)] \cdot \delta_{\mu^*(W)}(U)| dU\, dZ \qquad \text{(Using (21))} \\
&= \frac{1}{2} \int |[p(Z \mid \mu^*(W), x) - p(Z \mid W, x)] \cdot \delta_{\mu^*(W)}(U)| dU\, dZ \\
&\leq \frac{1}{2} \int |p(Z \mid \mu^*(W), x) - p(Z \mid W, x)| \cdot \delta_{\mu^*(W)}(U) dU\, dZ \\
&\leq \frac{1}{2} \int L_{Z,U,x} \cdot O(\sigma_\epsilon) \cdot \delta_{\mu^*(W)}(U) dU\, dZ \qquad \text{(Using (26))} \\
&= \frac{L_{Z,U,x}}{2} \cdot O(\sigma_\epsilon).
\end{aligned} \tag{28}$$

Applying reverse Pinsker's inequality[1], which states

$$D_{\mathrm{KL}}(P\|Q) \leq \frac{2\log e}{Q_{\min}} \mathrm{TV}(P, Q)^2 \tag{29}$$

where $Q_{\min} := \min_{(U,Z)} p(U \mid W, x) \cdot p(Z \mid W, x)$ and we assume $Q_{\min} > 0$. Combining (28) and (29), we can derive

$$D_{\mathrm{KL}}(P\|Q) \leq \frac{L_{Z,U,x}^2 \log e}{2Q_{\min}} O(\sigma_\epsilon^2). \tag{30}$$

Thus, for the mutual information $I(U; Z \mid W, x) = \mathbb{E}_{W|x} D_{\mathrm{KL}}(P\|Q)$, we have

$$I(U; Z \mid W, x) \leq \mathbb{E}_{W|x}\left[\frac{L_{Z,U,x}^2 \log e}{2Q_{\min}} O(\sigma_\epsilon^2)\right] = \frac{L_{Z,U,x}^2 \log e}{2Q_{\min}} O(\sigma_\epsilon^2). \tag{31}$$

Finally, by the definition $O(\cdot)$, we know that there exists there exists $C_0 > 0$ depending on $L_{Z,U,x}$ such that

$$I(U; Z|W, x) \leq C_0 \sigma_\epsilon^2$$

for sufficiently small $\sigma_\epsilon$. Hence, $I(U; Z|W, x) \to 0$ as $\sigma_\epsilon^2 \to 0$, completing the proof.

## C  Relative Error for Structural Equation Model (SEM)

Consider a causal graph in Figure 4 induced by the following structural equations model (SEM) [7]:

$$Y = \alpha_{YX} X + \alpha_{YW} W + \alpha_{YU} U + \varepsilon_Y \tag{32}$$
$$W = \alpha_{WU} U + \varepsilon_W \tag{33}$$
$$Z = \alpha_{ZU} U + \varepsilon_Z \tag{34}$$
$$X = \alpha_{XZ} Z + \alpha_{XU} U + \varepsilon_X , \tag{35}$$

where $\alpha_{YX}$, $\alpha_{YW}$, $\alpha_{YU}$, $\alpha_{ZU}$ are nonzero real coefficients, and $\varepsilon_Y \sim \mathcal{N}(0, \sigma_Y^2), \varepsilon_W \sim \mathcal{N}(0, \sigma_W^2), \varepsilon_Z \sim \mathcal{N}(0, \sigma_Z^2), \varepsilon_X \sim \mathcal{N}(0, \sigma_X^2)$, are all drawn from zero mean Gaussian distributions with distinct variance.

Consider the SEM in (32)-(35) with the unobserved confounder $U$ also assumed to be Gaussian, *i.e.*, $U \sim \mathcal{N}(0, \sigma_U^2)$. This is done for convenience, because in such a case, $W, Z$ and $X$ being linear in $U$, thus $(W, Z, X, U)$ jointly follow a multivariate Gaussian distribution. More specifically, since we know that $\mathbb{E}[U] = 0$,

$$\mathbb{E}[W] = \mathbb{E}[\alpha_{WU} U + \varepsilon_W] = \mathbb{E}[\alpha_{WU} U] = \alpha_{WU} \mathbb{E}[U] = 0 \tag{36}$$
$$\mathbb{E}[Z] = \mathbb{E}[\alpha_{ZU} U + \varepsilon_Z] = \mathbb{E}[\alpha_{ZU} U] = \alpha_{ZU} \mathbb{E}[U] = 0, \tag{37}$$

---

[1]This inequality is stated in Section 7.6 of [37].

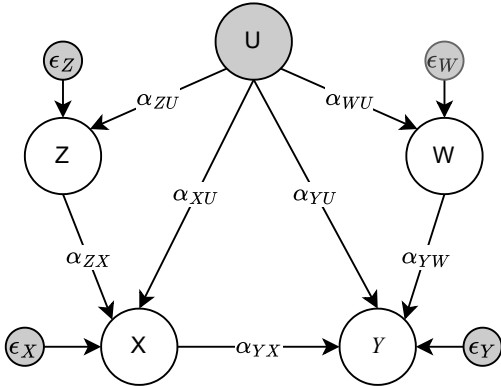

Figure 4: Graphical model of the structural equation model (SEM) in (32):(35). Grayed out nodes represent unobserved random variables.

and

$$\mathbb{E}[X] = \mathbb{E}[\alpha_{XZ}Z + \alpha_{XU}U + \varepsilon_X] = \mathbb{E}[\alpha_{XZ}Z + \alpha_{XU}U] = \alpha_{XZ}\mathbb{E}[Z] + \alpha_{XU}\mathbb{E}[U] = 0, \quad (38)$$

hence $(W, Z, X, U)$ can be expressed as $\mathcal{N}(0, \Sigma_{WZXU,WZXU})$, where $\Sigma_{WZXU,WZXU}$ is the corresponding covariance matrix whose entries are stated in the following proposition.

**Proposition 3** *Assume that the unobserved confounder $U$, treatment variable $X$, outcome variable $Y$, treatment-related proxy variable $Z$ and outcome-related variable $W$ satisfy the structural equation model and $U \sim \mathcal{N}(0, \sigma_U^2)$. The entries of the covariance matrix $\Sigma_{WZXU,WZXU}$ would be*

$$\begin{aligned}
\mathrm{Cov}(U, U) &= \sigma_U^2, \\
\mathrm{Cov}(W, W) &= \alpha_{WU}^2 \sigma_U^2 + \sigma_W^2, \\
\mathrm{Cov}(Z, Z) &= \alpha_{ZU}^2 \sigma_U^2 + \sigma_Z^2, \\
\mathrm{Cov}(X, X) &= (\alpha_{XZ}\alpha_{ZU} + \alpha_{XU})^2 \sigma_U^2 + \alpha_{XZ}^2 \sigma_Z^2 + \sigma_X^2, \\
\mathrm{Cov}(W, U) &= \alpha_{WU}\sigma_U^2, \\
\mathrm{Cov}(Z, U) &= \alpha_{ZU}\sigma_U^2, \\
\mathrm{Cov}(X, U) &= (\alpha_{XZ}\alpha_{ZU} + \alpha_{XU})\sigma_U^2, \\
\mathrm{Cov}(W, Z) &= \alpha_{WU}\alpha_{ZU}\sigma_U^2, \\
\mathrm{Cov}(W, X) &= \alpha_{WU}(\alpha_{XZ}\alpha_{ZU} + \alpha_{XU})\sigma_U^2, \\
\mathrm{Cov}(Z, X) &= \alpha_{ZU}(\alpha_{XZ}\alpha_{ZU} + \alpha_{XU})\sigma_U^2 + \alpha_{XZ}\sigma_Z^2.
\end{aligned}$$

**Proof:** In the derivations that follow, we will make use of the linearity of expectation and the formula for the variance of a linear combination of random variables.

Further, the derivations will repeatedly use the fact that, since, $U$, $\varepsilon_W$, $\varepsilon_Z$, and $\varepsilon_X$ are mutually independent,

$$\begin{aligned}
\mathbb{E}[U\varepsilon_Z] &= \mathbb{E}[U]\mathbb{E}[\varepsilon_Z] = 0, \\
\mathbb{E}[U\varepsilon_X] &= \mathbb{E}[U]\mathbb{E}[\varepsilon_X] = 0, \\
\mathbb{E}[\varepsilon_W U] &= \mathbb{E}[\varepsilon_W]\mathbb{E}[U] = 0, \\
\mathbb{E}[\varepsilon_W \varepsilon_Z] &= \mathbb{E}[\varepsilon_W]\mathbb{E}[\varepsilon_Z] = 0, \\
\mathbb{E}[\varepsilon_W \varepsilon_X] &= \mathbb{E}[\varepsilon_W]\mathbb{E}[\varepsilon_X] = 0.
\end{aligned}$$

- $\mathrm{Cov}(U, U)$. Following the definition, we have $\mathrm{Cov}(U, U) = \mathrm{Var}(U) = \sigma_U^2$.

- $\text{Cov}(W, W)$. Note that $W = \alpha_{WU}U + \varepsilon_W$. We can derive

$$\text{Cov}(W, W) = \text{Var}(W) = \text{Var}(\alpha_{WU}U + \varepsilon_W)$$
$$= \alpha_{WU}^2 \text{Var}(U) + \text{Var}(\varepsilon_W) \tag{39}$$
$$= \alpha_{WU}^2 \sigma_U^2 + \sigma_W^2. \tag{40}$$

(39) holds because $U \perp\!\!\!\perp \varepsilon_W$.

- $\text{Cov}(Z, Z)$. Note that $Z = \alpha_{ZU}U + \varepsilon_Z$. We can derive

$$\text{Cov}(Z, Z) = \text{Var}(Z) = \text{Var}(\alpha_{ZU}U + \varepsilon_Z)$$
$$= \alpha_{ZU}^2 \text{Var}(U) + \text{Var}(\varepsilon_Z) \tag{41}$$
$$= \alpha_{ZU}^2 \sigma_U^2 + \sigma_Z^2. \tag{42}$$

(41) uses the fact that $U \perp\!\!\!\perp \varepsilon_z$.

- $\text{Cov}(X, X)$. Note that $X = \alpha_{XZ}Z + \alpha_{XU}U + \varepsilon_X$. We have

$$\text{Cov}(X, X) = \text{Var}(X) = \text{Var}(\alpha_{XZ}Z + \alpha_{XU}U + \varepsilon_X)$$
$$= \text{Var}(\alpha_{XZ}[\alpha_{ZU}U + \varepsilon_Z] + \alpha_{XU}U + \varepsilon_X)$$
$$= \text{Var}((\alpha_{XZ}\alpha_{ZU} + \alpha_{XU})U + \alpha_{XZ}\varepsilon_Z + \varepsilon_X)$$
$$= (\alpha_{XZ}\alpha_{ZU} + \alpha_{XU})^2\text{Var}(U) + \alpha_{XZ}^2\text{Var}(\varepsilon_Z) + \text{Var}(\varepsilon_X)] \tag{43}$$
$$= [\alpha_{XZ}\alpha_{ZU} + \alpha_{XU}]^2\sigma_U^2 + \alpha_{XZ}^2\sigma_Z^2 + \sigma_X^2. \tag{44}$$

(43) follows from the fact that $U$, $\varepsilon_Z$ and $\varepsilon_X$ are mutually independent.

- $\text{Cov}(W, U)$. Note that $\text{Cov}(W, U) = \mathbb{E}[W \cdot U] - \mathbb{E}[W] \cdot \mathbb{E}[U]$. We have

$$\mathbb{E}[W \cdot U] = \mathbb{E}[(\alpha_{WU}U + \varepsilon_W) \cdot U] = \mathbb{E}[\alpha_{WU}U^2 + \varepsilon_W U]$$
$$= \alpha_{WU}\mathbb{E}[U^2] + \mathbb{E}[\varepsilon_W U]$$
$$= \alpha_{WU}\mathbb{E}[U^2] = \alpha_{WU}(\text{Var}(U) + \mathbb{E}[U]^2) = \alpha_{WU}\sigma_U^2, \tag{45}$$

and,

$$\mathbb{E}[W] \cdot \mathbb{E}[U] = 0 \cdot 0 = 0.$$

Thus, we can infer that $\text{Cov}(W, U) = \alpha_{WU}\sigma_U^2$.

- $\text{Cov}(Z, U)$. Note that $\text{Cov}(Z, U) = \mathbb{E}[Z \cdot U] - \mathbb{E}[Z] \cdot \mathbb{E}[U]$. We have

$$\mathbb{E}[Z \cdot U] = \mathbb{E}[(\alpha_{ZU}U + \varepsilon_Z) \cdot U] = \mathbb{E}[\alpha_{ZU}U^2 + U\varepsilon_Z]$$
$$= \alpha_{ZU}\mathbb{E}[U^2] + \mathbb{E}[U\varepsilon_Z] \tag{46}$$
$$= \alpha_{ZU}\mathbb{E}[U^2] = \alpha_{ZU}(\text{Var}(U) + \mathbb{E}[U]^2) = \alpha_{ZU}\sigma_U^2, \tag{47}$$

and,

$$\mathbb{E}[Z] \cdot \mathbb{E}[U] = 0 \cdot 0 = 0,$$

we can infer that $\text{Cov}(Z, U) = \alpha_{ZU}\sigma_U^2$.

- $\text{Cov}(X, U)$. Note that $\text{Cov}(X, U) = \mathbb{E}[X \cdot U] - \mathbb{E}[X] \cdot \mathbb{E}[U]$. Since we have

$$\mathbb{E}[X \cdot U] = \mathbb{E}[(\alpha_{XZ}Z + \alpha_{XU}U + \varepsilon_X) \cdot U]$$
$$= \mathbb{E}[((\alpha_{XZ}\alpha_{ZU} + \alpha_{XU})U + \alpha_{XZ}\varepsilon_Z + \varepsilon_X) \cdot U]$$
$$= (\alpha_{XZ}\alpha_{ZU} + \alpha_{XU})\mathbb{E}[U^2] + \alpha_{XZ}\mathbb{E}[\varepsilon_Z U] + \mathbb{E}[\varepsilon_X U]$$
$$= (\alpha_{XZ}\alpha_{ZU} + \alpha_{XU})\mathbb{E}[U^2] + \alpha_{XZ}\mathbb{E}[\varepsilon_Z]\mathbb{E}[U] + \mathbb{E}[\varepsilon_X]\mathbb{E}[U] \tag{48}$$
$$= (\alpha_{XZ}\alpha_{ZU} + \alpha_{XU})\mathbb{E}[U^2]$$
$$= (\alpha_{XZ}\alpha_{ZU} + \alpha_{XU})(\text{Var}(U) + \mathbb{E}[U]^2)$$
$$= (\alpha_{XZ}\alpha_{ZU} + \alpha_{XU})\sigma_U^2, \tag{49}$$

and,

$$\mathbb{E}[X] \cdot \mathbb{E}[U] = 0 \cdot 0 = 0,$$

we can infer that $\text{Cov}(X, U) = (\alpha_{XZ}\alpha_{ZU} + \alpha_{XU})\sigma_U^2$.
(48) uses the fact that $U \perp\!\!\!\perp \varepsilon_Z$ and $U \perp\!\!\!\perp \varepsilon_X$.

- $\text{Cov}(W, Z)$. Note that $\text{Cov}(W, Z) = \mathbb{E}[W \cdot Z] - \mathbb{E}[W] \cdot \mathbb{E}[Z]$. Since we can derive,

$$
\begin{aligned}
\mathbb{E}[W \cdot Z] &= \mathbb{E}[(\alpha_{WU}U + \varepsilon_W) \cdot (\alpha_{ZU}U + \varepsilon_Z)] \\
&= \mathbb{E}[(\alpha_{WU}\alpha_{ZU}U^2 + \varepsilon_W \alpha_{ZU}U + \alpha_{WU}U\varepsilon_Z + \varepsilon_W \varepsilon_Z] \\
&= \mathbb{E}[\alpha_{WU}\alpha_{ZU}U^2] + \mathbb{E}[\varepsilon_W \alpha_{ZU}U] + \mathbb{E}[\alpha_{WU}U\varepsilon_Z] + \mathbb{E}[\varepsilon_W \varepsilon_Z] \\
&= \mathbb{E}[\alpha_{WU}\alpha_{ZU}U^2] + \alpha_{ZU}\mathbb{E}[\varepsilon_W]\mathbb{E}[U] + \alpha_{WU}\mathbb{E}[U]\mathbb{E}[\varepsilon_Z] + \mathbb{E}[\varepsilon_W]\mathbb{E}[\varepsilon_Z] \quad (50) \\
&= \mathbb{E}[\alpha_{WU}\alpha_{ZU}U^2] \\
&= \alpha_{WU}\alpha_{ZU}\mathbb{E}[U^2] = \alpha_{WU}\alpha_{ZU}(\text{Var}(U) + \mathbb{E}[U]^2) = \alpha_{WU}\alpha_{ZU}\sigma_U^2, \quad (51)
\end{aligned}
$$

and

$$
\mathbb{E}[W] \cdot \mathbb{E}[Z] = 0 \cdot 0 = 0,
$$

we can infer that $\text{Cov}(W, Z) = \alpha_{WU}\alpha_{ZU}\sigma_U^2$.
(50) follows from the fact that using $U$, $\varepsilon_W$ and $\varepsilon_Z$ are mutually independent.

- $\text{Cov}(W, X)$. We can derive

$$
\begin{aligned}
\mathbb{E}[W \cdot X] &= \mathbb{E}[(\alpha_{WU}U + \varepsilon_W) \cdot (\alpha_{XZ}Z + \alpha_{XU}U + \varepsilon_X)] \\
&= \mathbb{E}[(\alpha_{WU}U + \varepsilon_W) \cdot (\alpha_{XZ}(\alpha_{ZU}U + \varepsilon_Z) + \alpha_{XU}U + \varepsilon_X)] \\
&= \mathbb{E}[(\alpha_{WU}U + \varepsilon_W) \cdot ((\alpha_{XZ}\alpha_{ZU} + \alpha_{XU})U + \alpha_{XZ}\varepsilon_Z + \varepsilon_X)] \\
&= \mathbb{E}[\alpha_{WU}(\alpha_{XZ}\alpha_{ZU} + \alpha_{XU})U^2 + \alpha_{WU}\alpha_{XZ}U\varepsilon_Z + \alpha_{WU}U\varepsilon_X \\
&\quad + (\alpha_{XZ}\alpha_{ZU} + \alpha_{XU})\varepsilon_W U + \alpha_{XZ}\varepsilon_W \varepsilon_Z + \varepsilon_W \varepsilon_X] \\
&= \alpha_{WU}(\alpha_{XZ}\alpha_{ZU} + \alpha_{XU})\mathbb{E}[U^2] + \alpha_{WU}\alpha_{XZ}\mathbb{E}[U\varepsilon_Z] + \alpha_{WU}\mathbb{E}[U\varepsilon_X] \\
&\quad + (\alpha_{XZ}\alpha_{ZU} + \alpha_{XU})\mathbb{E}[\varepsilon_W U] + \alpha_{XZ}\mathbb{E}[\varepsilon_W \varepsilon_Z] + \mathbb{E}[\varepsilon_W \varepsilon_X] \\
&= \mathbb{E}[(\alpha_{WU}(\alpha_{XZ}\alpha_{ZU} + \alpha_{XU})U^2] \\
&= \alpha_{WU}(\alpha_{XZ}\alpha_{ZU} + \alpha_{XU})\mathbb{E}[U^2] \\
&= \alpha_{WU}(\alpha_{XZ}\alpha_{ZU} + \alpha_{XU})(\text{Var}(U^2) - \mathbb{E}[U]^2) \\
&= \alpha_{WU}(\alpha_{XZ}\alpha_{ZU} + \alpha_{XU})\sigma_U^2. \quad (52)
\end{aligned}
$$

Besides, since $\mathbb{E}[W] \cdot \mathbb{E}[X] = 0$, we can infer that $\text{Cov}(W, X) = \alpha_{WU}(\alpha_{XZ}\alpha_{ZU} + \alpha_{XU})\sigma_U^2$.

- $\text{Cov}(Z, X)$. Finally, we consider that

$$
\begin{aligned}
\mathbb{E}[Z \cdot X] &= \mathbb{E}[(\alpha_{ZU}U + \varepsilon_Z) \cdot (\alpha_{XZ}Z + \alpha_{XU}U + \varepsilon_X)] \\
&= \mathbb{E}[(\alpha_{ZU}U + \varepsilon_Z) \cdot (\alpha_{XZ}(\alpha_{ZU}U + \varepsilon_Z) + \alpha_{XU}U + \varepsilon_X)] \\
&= \mathbb{E}[(\alpha_{ZU}U + \varepsilon_Z) \cdot ((\alpha_{XZ}\alpha_{ZU} + \alpha_{XU})U + \alpha_{XZ}\varepsilon_Z + \varepsilon_X)] \\
&= \mathbb{E}[\alpha_{ZU}(\alpha_{XZ}\alpha_{ZU} + \alpha_{XU})U^2 + \alpha_{ZU}\alpha_{XZ}U\varepsilon_Z + \alpha_{ZU}U\varepsilon_X \\
&\quad + (\alpha_{XZ}\alpha_{ZU} + \alpha_{XU})U\varepsilon_Z + \alpha_{XZ}\varepsilon_Z^2 + \varepsilon_Z \varepsilon_X] \\
&= \alpha_{ZU}(\alpha_{XZ}\alpha_{ZU} + \alpha_{XU})\mathbb{E}[U^2] + \alpha_{ZU}\alpha_{XZ}\mathbb{E}[U\varepsilon_Z] + \alpha_{ZU}\mathbb{E}[U\varepsilon_X] \\
&\quad + (\alpha_{XZ}\alpha_{ZU} + \alpha_{XU})\mathbb{E}[U\varepsilon_Z] + \alpha_{XZ}\mathbb{E}[\varepsilon_Z^2] + \mathbb{E}[\varepsilon_Z \varepsilon_X] \\
&= \alpha_{ZU}(\alpha_{XZ}\alpha_{ZU} + \alpha_{XU})\mathbb{E}[U^2] + \alpha_{XZ}\mathbb{E}[\varepsilon_Z^2] \quad (53) \\
&= \alpha_{ZU}(\alpha_{XZ}\alpha_{ZU} + \alpha_{XU})(\text{Var}(U^2) - \mathbb{E}[U]^2) + \alpha_{XZ}(\text{Var}(\varepsilon_Z) + \mathbb{E}[\varepsilon_Z]^2) \\
&= \alpha_{ZU}(\alpha_{XZ}\alpha_{ZU} + \alpha_{XU})\sigma_U^2 + \alpha_{XZ}\sigma_Z^2 \quad (54)
\end{aligned}
$$

Moreover, since $\mathbb{E}[Z] \cdot \mathbb{E}[X] = 0$, we can infer that $\text{Cov}(Z, X) = \alpha_{ZU}(\alpha_{XZ}\alpha_{ZU} + \alpha_{XU})\sigma_U^2 + \alpha_{XZ}\sigma_Z^2$.

$\blacksquare$

**Closed-from Expression for Relative Approximation Error**  With the help of Proposition 3, we can start to derive the closed-from expression for the relative error $r(\eta_i)$. Given a pair of Gaussian random vectors,

$$
(A, B)^T \sim \mathcal{N}\left(0, \begin{pmatrix} \Sigma_{A,A} & \Sigma_{A,B} \\ \Sigma_{B,A} & \Sigma_{B,B} \end{pmatrix}\right),
$$

the conditional $A \mid B = b$ can be expressed as

$$A \mid B = b \sim \mathcal{N}(\Sigma_{A,B}\Sigma_{B,B}^{-1}b, \ \Sigma_{A,A} - \Sigma_{A,B}\Sigma_{B,B}^{-1}\Sigma_{B,A}), \tag{55}$$

as stated in section 2.3.1 of [6].

With this expression and Proposition 3, we can express $p(W|X, Z)$, $p(U|W, Z, X)$ and $p(U|W, X)$ as Gaussian distributions using components of $\Sigma_{WZXU}$ as follows.

- $p(W \mid X, Z)$. Let $A = W$, $B = (X, Z)^T$ and $(X = x, Z = z)$. Then, we have

$$\mu_{W|(X=x,Z=z)} = \Sigma_{W,XZ}\Sigma_{XZ,XZ}^{-1} \begin{pmatrix} x \\ z \end{pmatrix}, \tag{56}$$

$$\sigma_{W|(X=x,Z=z)}^2 = \sigma_W^2 - \Sigma_{W,XZ}\Sigma_{XZ,XZ}^{-1}\Sigma_{XZ,W}. \tag{57}$$

- $p(U \mid W, X, Z)$. Let $A = U$, $B = (W, X, Z)^T$ and $(W = w, X = x, Z = z)$. Then,

$$\mu_{U|(W=w,X=x,Z=z)} = \Sigma_{U,WXZ}\Sigma_{WXZ,WXZ}^{-1} \begin{pmatrix} w \\ x \\ z \end{pmatrix}, \tag{58}$$

$$\sigma_{U|(W=w,X=x,Z=z)}^2 = \sigma_U^2 - \Sigma_{U,WXZ}\Sigma_{WXZ,WXZ}^{-1}\Sigma_{WXZ,U}. \tag{59}$$

- $p(U \mid W, X)$. Let $A = U$, $B = (W, X)^T$ and $(W = w, X = x)$. Similarly, we have

$$\mu_{U|(W=w,X=x)} = \Sigma_{U,WX}\Sigma_{WX,WX}^{-1} \begin{pmatrix} w \\ x \end{pmatrix}, \tag{60}$$

$$\sigma_{U|(W=w,X=x)}^2 = \sigma_U^2 - \Sigma_{U,WX}\Sigma_{WX,WX}^{-1}\Sigma_{WX,U}. \tag{61}$$

Recall that $\frac{\eta_i}{|\mathbb{E}[Y|x_i,z_i]|}$ for the SEM, *before* the expectation wrt $p(Z|X)$ and $p(X)$ can be expressed (in a slight abuse of notation) as

$$\begin{aligned}
\frac{\eta_i}{|\mathbb{E}[Y|x_i,z_i]|} &= \frac{|\mathbb{E}[Y|x_i,z_i] - \mathbb{E}_{W\sim p(W|x_i,z_i)}[b(W,x_i)]|}{|\mathbb{E}[Y|x_i,z_i]|} \\
&= \frac{\left|\alpha_{YU}\,\mathbb{E}_{W|x_i,z_i}\left[\mathbb{E}_{U|W,x_i,z_i}[U] - \mathbb{E}_{U|W,x_i}[U]\right]\right|}{\left|\alpha_{YX}x_i + \mathbb{E}_{W|x_i,z_i}\left[\alpha_{YW}W + \alpha_{YU}\,\mathbb{E}_{U|W,x_i,z_i}[U]\right]\right|}.
\end{aligned} \tag{62}$$

With (56)-(61), we can further express $\frac{\eta_i}{|\mathbb{E}[Y|x_i,z_i]|}$ as

$$\frac{\eta_i}{|\mathbb{E}[Y|x_i,z_i]|} = \frac{|\alpha_{YU}[\mu_{U|(W=w^*,X=x_i,Z=z_i)} - \mu_{U|(W=w^*,X=x_i)}]|}{|\alpha_{YX}x_i + \alpha_{YW}w^* + \mu_{U|(W=w^*,X=x_i,Z=z_i)}|}. \tag{63}$$

where $w^* = \mu_{W|(X=x_i,Z=z_i)}$. Thus, we can expressed $r(\eta_i)$ as

$$\begin{aligned}
r(\eta_i) &= \mathbb{E}_{z_i\sim p(Z|X=x_i)}\left[\frac{\eta_i}{|\mathbb{E}[Y|x_i,z_i]|}\right] \\
&= \mathbb{E}_{z_i\sim p(Z|X=x_i)}\left[\frac{|\alpha_{YU}[\mu_{U|(W=w^*,X=x_i,Z=z_i)} - \mu_{U|(W=w^*,X=x_i)}]|}{|\alpha_{YX}x_i + \alpha_{YW}w^* + \mu_{U|(W=w^*,X=x_i,Z=z_i)}|}\right].
\end{aligned} \tag{64}$$

**Analytic Solution for $r(\eta) = 0$** From (64), we note that if we can show $\mu_{U|(W=w,X=x,Z=z)} = \mu_{U|(W=w,X=x)}$ for any $(W = w, X = x, Z = z)$, then $r(\eta_i)$ would be zero. It is clear that when $U \perp\!\!\!\perp Z|(W, X)$, then $\mu_{U|(W,X,Z)} = \mu_{U|(W,X)}$ holds, and hence $r(\eta_i)$ would be zero. Following this idea, we can derive a sufficient condition such that $r(\eta_i) = 0$.

**Lemma 1** *Assume that the unobserved confounder $U$, treatment variable $X$, outcome variable $Y$, treatment-related proxy variable $Z$ and outcome-related variable $W$ satisfy the SEM and $U \sim \mathcal{N}(0, \sigma_U^2)$. Then, if*

$$\frac{\sigma_X^2}{\sigma_Z^2} = \frac{\alpha_{XU}\alpha_{XZ}}{\alpha_{ZU}},$$

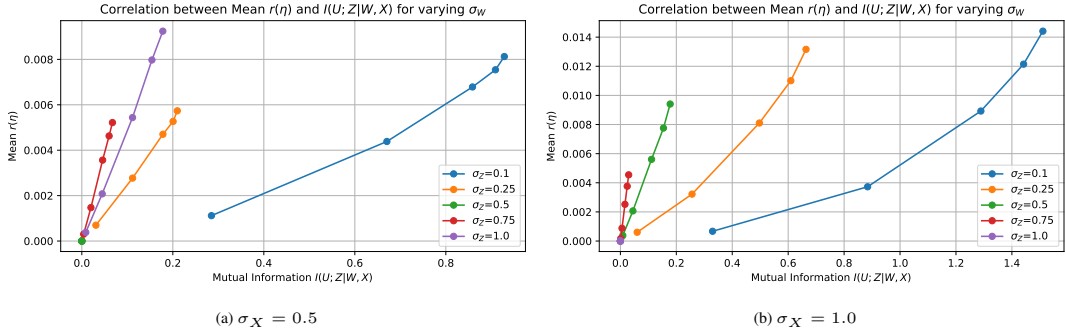

(a) $\sigma_X = 0.5$        (b) $\sigma_X = 1.0$

Figure 5: Relative approximation error ($r(\eta)$) *vs.* mutual information ($I(U; Z|W, x)$) both averaged over $X$. Each line represents a value of $\sigma_Z$ for increasing values of $\sigma_W = \{0.1, 0.25, 0.5, 0.75, 1\}$, which are consistent with $I(U; Z|W, x)$.

*then we can infer that $U \perp\!\!\!\perp Z|(W, X)$ and hence $r(\eta) = 0$.*

**Proof:** By (55), we can derive

$$\Sigma_{UZ|WX,UZ|WX} = \Sigma_{UZ,UZ} - \Sigma_{UZ,WX}\Sigma_{WX,WX}^{-1}\Sigma_{WX,UZ}, \tag{65}$$

which implies that

$$\text{Cov}(U, Z \mid W, X) = \text{Cov}(U, Z) - \Sigma_{U,WX}\Sigma_{WX,WX}^{-1}\Sigma_{WX,Z} \tag{66}$$

Finally, we plug-in the value of Proposition 3 into (66). Thus, we can derive

$$\begin{aligned}
&\text{Cov}(U, Z \mid W, X) \\
&= \frac{\sigma_U^2\,\sigma_W^2(-\alpha_{XU}\,\alpha_{XZ}\,\sigma_Z^2 + \alpha_{ZU}\,\sigma_X^2)}{\alpha_{WU}^2\alpha_{XZ}^2\,\sigma_U^2\sigma_Z^2 + \alpha_{WU}^2\,\sigma_U^2\sigma_X^2 + \alpha_{XU}^2\,\sigma_U^2\sigma_W^2 + 2\,\alpha_{XU}\alpha_{XZ}\alpha_{ZU}\,\sigma_U^2\sigma_W^2 + \alpha_{XZ}^2\alpha_{ZU}^2\,\sigma_U^2\sigma_W^2 + \alpha_{XZ}^2\,\sigma_W^2\sigma_Z^2 + \sigma_W^2\sigma_X^2}
\end{aligned} \tag{67}$$

Assume $\frac{\sigma_X^2}{\sigma_Z^2} = \frac{\alpha_{XU}\alpha_{XZ}}{\alpha_{ZU}}$. Then, $\sigma_U^2\sigma_W^2(-\alpha_{XU}\alpha_{XZ}\sigma_Z^2 + \alpha_{ZU}\sigma_X^2) = 0$ and hence $\text{Cov}(U, Z \mid W, X) = 0$, which suggests that $U$ and $Z$ are uncorrelated conditioned on $W, X$.

Further, since $(U, Z)|(W, X)$ are jointly Gaussian, uncorrelatedness implies independence, as stated in section 11.5 of [26].

Hence, we can conclude that if $\sigma_U^2\sigma_W^2(-\alpha_{XU}\alpha_{XZ}\sigma_Z^2 + \alpha_{ZU}\sigma_X^2) = 0$, $U \perp\!\!\!\perp Z|(W, X)$ and, thus $r(\eta_i) = 0$. $\blacksquare$

Lemma 1 seems to indicate that once the treatment variable $X$ and treatment-related proxy variable $Z$ satisfy some relation, then the information of $U$ in $Z$ would be contained in the information of $U$ in $(W, X)$. In other words, we can have an approximation of $\mathbb{E}[U|W, X, Z]$ with $\mathbb{E}[U|W, X]$, *i.e.*, $\mathbb{E}[U|W, X, Z] \approx \mathbb{E}[U|W, X]$, which implies that $r(\eta)$ would be close to zero.

## D    SEM Experiments for Gaussian Unobserved Confounder

To analyze the association between the relative error $r(\eta) = \mathbb{E}_{Z \sim p(Z|x)}[\eta/|\mathbb{E}[Y|x, Z]|]$ *vs.* $I(U; Z|W, x)$ both averaged over $X$, we present Figure 2 showing the results for the mean of the relative error $r(\eta)$ *vs.* $I(U; Z|W, x)$ averaged over $X$, for $\sigma_U = 10$, $\sigma_X = 0.1$ and $\sigma_Z, \sigma_W = \{0.1, 0.25, 0.5, 0.75, 1\}$, all weights $\alpha_{YX}$, $\alpha_{YW}$, $\alpha_{YU}$, $\alpha_{ZU}$ are set to 1. Moreover, to show the effect of changing $\sigma_X$, we set $\sigma_X = 0.5$ and $\sigma_X = 1.0$ in Figure 5(a) and Figure 5(b), respectively. For these experiments, we generate $M = 10,000$ samples, $\mathcal{D} = \{(u_i, w_i, z_i\}_{i=1,M}$, drawn using $U \sim \mathcal{N}(0, \sigma_U^2)$ and (34)-(35), and the corresponding noise variances to compute the mean of the relative error $r(\eta)$ and mutual information $I(U; Z|W, x)$ averaged over $X$.

We observe that across $\sigma_X = \{0.1, 0.5, 1.0\}$, increasing $\sigma_W$ drives up both the mutual information $I(U; Z|W, x)$ and the relative error $r(\eta)$ in tandem. This shows that our mutual information-based bound reliably captures the growth of the error with increasing $\sigma_W$. Furthermore, we present the

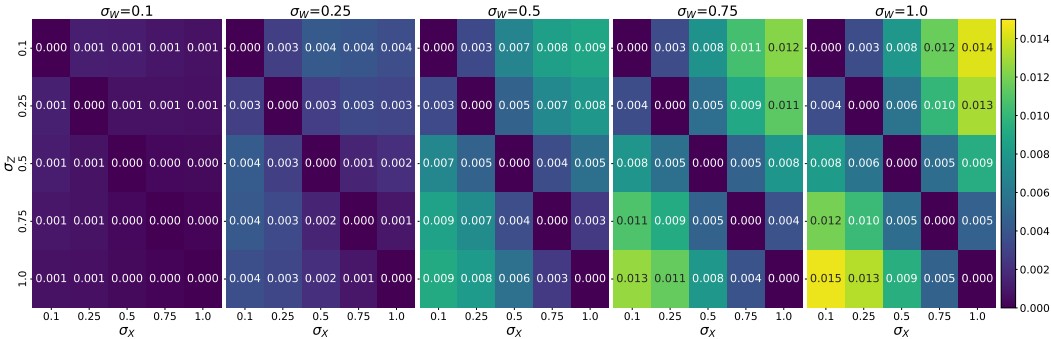

Figure 6: Heatmap of Mean of $\frac{\eta}{|\mathbb{E}[Y|x,Z]|}$

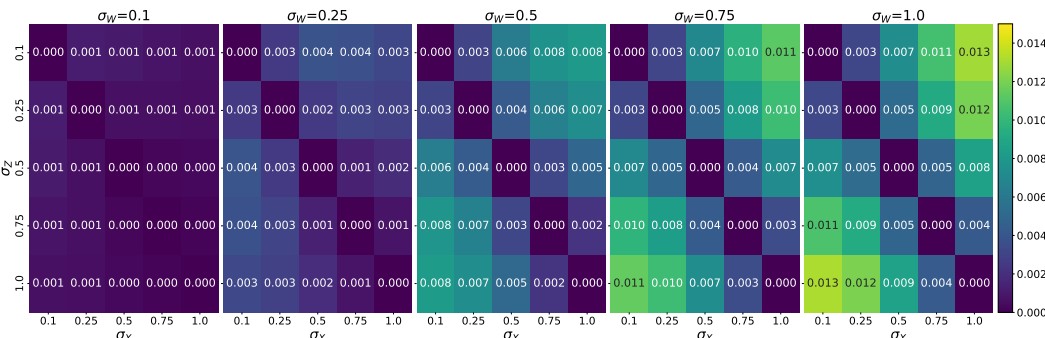

Figure 7: Heatmap of Standard Deviation of $\frac{\eta}{|\mathbb{E}[Y|x,Z]|}$

heatmap of the mean and standard deviation of $\frac{\eta}{|\mathbb{E}[Y|x,Z]|}$ in Figure 6 and Figure 7, respectively. It is worth mentioning that the mean of $\frac{\eta}{|\mathbb{E}[Y|x,Z]|}$ is equal to the mean of $r(\eta)$, and that the standard deviation of $\frac{\eta}{|\mathbb{E}[Y|x,Z]|}$ is actually the upper bound of the standard deviation of the relative error $r(\eta)$, which is induced by the Jensen's inequality. We observe that the mean of $r(\eta)$ and standard deviation of $\frac{\eta}{|\mathbb{E}[Y|x,Z]|}$ monotonically increase as $\sigma_W$ increases. Notably, when $\sigma_X = \sigma_Z$, both $r(\eta) = 0$ and $I(U; Z|W, x) = 0$, which is a special case formalized in Lemma 1 in Appendix C.

Note that when computing the statistics of the relative error, we use trimmed mean and standard deviation, *i.e.*, we remove $10\%$ of the outlying relative error values to mitigate the influence of large outliers occurring when the denominator of $r(\eta)$ is too small, thus causing precision errors.

# E  Experimental Details

In both experiments, we generate $M = 1000$ or $5000$ data for both stage $\mathcal{D}_1 = \{(x_i, z_i, w_i)\}_{i=1,M}$ and $\mathcal{D}_2 = \{(x_i, z_i, y_i)\}_{i=1,M}$ to share.

## E.1  Details of the Generation of the Demand Dataset

The observations are generated using the following model:

$$\epsilon \sim \mathcal{N}(0, 1)$$
$$Y = P\left(\exp\left(\frac{V - P}{10} \wedge 5\right) - 5g(D)\right) + \epsilon, \tag{68}$$

where $Y$ represents sales, $P$ is the treatment variable (price), and these are influenced by potential demand $D$. Here, $a \wedge b$ denotes $\min(a, b)$, and the function $g$ is defined as:

$$g(d) = 2\left(\frac{(d-5)^4}{600} + \exp(-(4(d-5)^2))\right) + \frac{d}{10} - 2.$$

As negative control (proxy) data, cost-shifters $C_1$ and $C_2$ are introduced as treatment-inducing proxies, and the views $V$ of the reservation page are used as the negative control outcome proxy data. The data is generated as follows:

$$
\begin{aligned}
D &\sim \text{Unif}[0, 10] \\
C_1 &\sim 2\sin(2D\pi/10) + \epsilon_1 \\
C_2 &\sim 2\cos(2D\pi/10) + \epsilon_2 \\
V &\sim 7g(D) + 45 + \epsilon_3 \\
P &= 35 + (C_1 + 3)g(D) + C_2 + \epsilon_4,
\end{aligned}
$$

where $\epsilon_1, \epsilon_2, \epsilon_3, \epsilon_4 \sim \mathcal{N}(0, 1)$. These specifications and notation are as given in [52]. Concerning the notation in this paper, we make the following correspondences: $D \leftrightarrow U$, $(C_1, C_2) \leftrightarrow Z$, $V \leftrightarrow W$, and $P \leftrightarrow X$. The outcome is $Y$, consistent with the notation of the main body of the paper. We select 10 evenly spaced treatment values within the range $[10, 30]$ as the test data, following the same setup introduced by [52].

## E.2   Details of the Generation of the dSprite dataset

This dataset consists of images parameterized by five variables (shape, scale, rotation, posX, and posY). The images are $64 \times 64$ pixels, resulting in a 4096-dimensional vector. The shape parameter is fixed to a "heart," hence using only heart-shaped images (see below). The other parameters take values within the following ranges: scale $\in [0.5, 1]$, rotation $\in [0, 2\pi]$, posX $\in [0, 1]$, and posY $\in [0, 1]$.

The treatment variable $A$ and the outcome $Y$ are generated as follows:

1. Uniformly sample latent parameters (scale, rotation, posX, posY).
2. Generate the treatment variable $A$ as

$$
A = \text{Fig}(\text{scale}, \text{rotation}, \text{posX}, \text{posY}) + \eta_A.
$$

3. Generate the outcome variable $Y$ as

$$
\begin{aligned}
\epsilon &\sim \mathcal{N}(0, 0.5) \\
Y &= \frac{1}{12}(\text{posY} - 0.5)\frac{(\text{vec}(B)^\top A)^2 - 3000}{500} + \epsilon.
\end{aligned}
$$

The function Fig returns the corresponding image for the latent parameters, and $\eta_A$ and $\epsilon$ are noise variables generated from $\eta_A \sim \mathcal{N}(0.0, 0.1I)$ and $\epsilon \sim \mathcal{N}(0, 0.5)$. The matrix $B \in \mathbb{R}^{64 \times 64}$ is defined as $B_{ij} = |32 - j|/32$. From this data generation process, we observe that $A$ and $Y$ are confounded by posY. While the treatment variable $A$ is given as a figure corrupted with Gaussian random noise, the variable posY is not revealed to the model, and hence there is no observable confounder.

The structural function $f_{\text{struct}}$ is defined as:

$$
f_{\text{struct}}(A) = \frac{(\text{vec}(B)^\top A)^2 - 3000}{500}.
$$

The negative control treatment is given by the tuple (scale, rotation, posX) $\in \mathbb{R}^3$, and the negative control outcome is

$$
\begin{aligned}
\eta_W &\sim \mathcal{N}(0.0, 0.1I) \\
W &= \text{Fig}(0.8, 0, 0.5, \text{posY}) + \eta_W.
\end{aligned}
$$

We set aside 588 test points to quantify the validation error (out of sample), which is generated from a grid of points on the latent variable space. The grids consist of 7 evenly spaced values for posX and posY, 3 evenly spaced values for scale, and 4 evenly spaced values for orientation. The above settings are as in [53].

Note that in [52], the dSprite data generation process and results are slightly different to those presented here. The experiment process was refined in [53] and is the one used in our experiments.

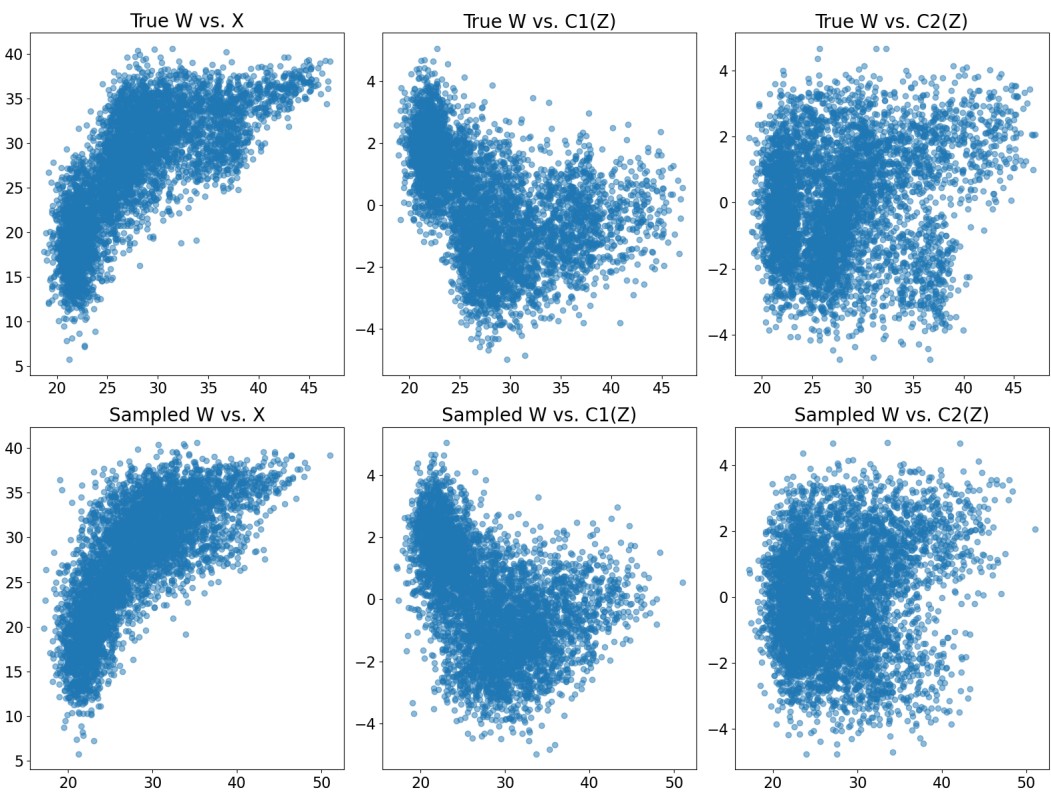

Figure 8: Samples drawn from the model and ground-truth distribution of $p(W|z, x)$ where $(Z, X)$ are drawn from the true distribution, and $Z = (C1, C2)$. From left to right in a row, samples are shown as $(w_i, x_i)$, $(w_i, C1_i)$ and $(w_i, C2_i)$. The top row is from the ground truth distribution and the bottom is from the model.

### E.3 Proxy Bucketing Strategy

Since the covariates in the Framingham dataset are not segregated into outcome-inducing proxy $W$ and the treatment-inducing proxy $Z$, we follow [47] to group the 32 covariates, namely: `age6`, `age61`, `age62`, `ascvd_hx6`, `bmi6`, `bmi61`, `bmi62`, `bpmeds6`, `chol5`, `chol51`, `chol52`, `dbp6`, `dbp61`, `dbp62`, `diab6`, `female`, `gluc5`, `gluc51`, `gluc52`, `hdl5`, `hdl51`, `hdl52`, `pad_hx6`, `sbp6`, `sbp61`, `sbp62`, `smoke6`, `stk_hx6`, `mi_hx6`, `trigly5`, `trigly51`, `trigly52`.

As we do not have the covariate groupings beforehand, we use $C$ to denote all covariates $(W, Z)$ together. We begin by ranking the proxies based on their strength of association with: $i$) The outcome from the coefficients obtained using CoxPH regression of $(Y, E)$ given $(X, C)$, and $ii$) The treatment from the coefficients obtained using logistic regression of $X$ given $C$. Then, we select proxies in decreasing order of strength of association, first choosing the proxy having the strongest correlation with the outcome as an outcome-inducing proxy, and we do the same for the treatment-inducing proxy. The algorithm halts upon allocation of all the proxies. We obtain the following proxy allocation (with 16 variables in both $W$ and $Z$):
$W$: `smoke6`, `female`, `ascvd_hx6`, `diab6`, `pad_hx6`, `gluc52`, `stk_hx6`, `age62`, `sbp62`, `gluc51`, `age61`, `age6`, `hdl5`, `gluc5`, `trigly52`, `trigly51`
$Z$: `mi_hx6`, `dbp62`, `bpmeds6`, `hdl52`, `dbp61`, `bmi62`, `hdl51`, `bmi61`, `bmi6`, `chol52`, `dbp6`, `sbp61`, `chol51`, `chol5`, `sbp6`, `trigly5`

### E.4 Model of $p(W|z, x)$ for Demand data

In Figure 8 we compare samples drawn from the ground-truth distribution of $p(W|z, x)$ to those of the model $p_\psi(W|Z = z, X = x)$, which is represented by a Gaussian distribution with mean and variance modeled via simple neural networks. We observe a close agreement between the true samples and our conditional distribution model.

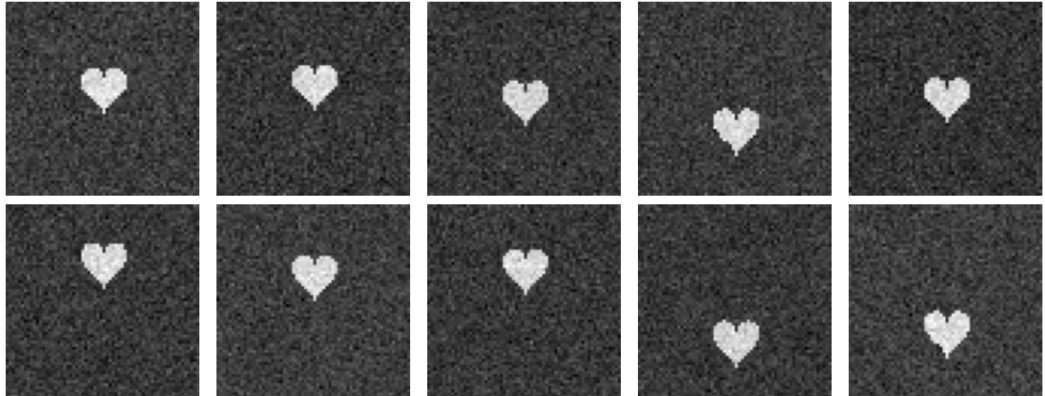

Figure 9: Examples of true dSprite proxy data $W$ (images).

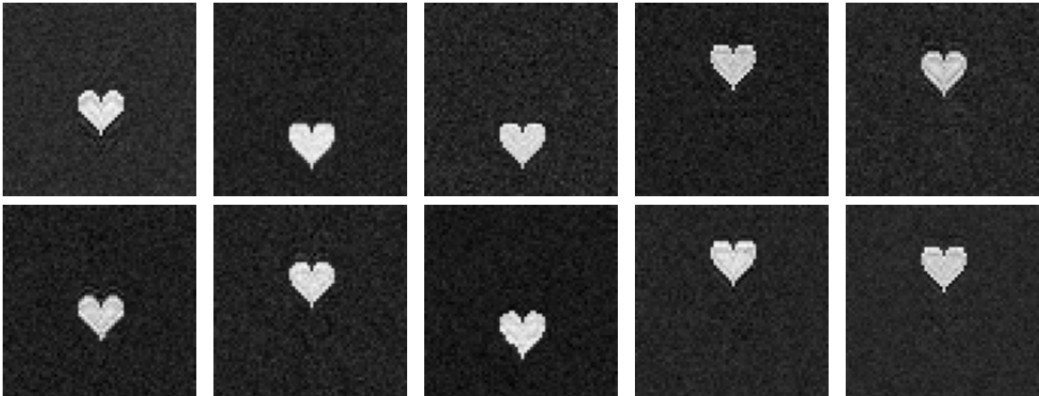

Figure 10: Examples of synthesized samples from $p(W|Z = z, X = x)$ for dSprite, manifested manifested via conditional GAN [32].

### E.5 Model of $p(W|z, x)$ for dSprite data

The capacity to sample from $p(W|z, x)$ for the dSprite data is implemented via conditional GAN [32]. Here, the proxy variable $W$ and the treatment $X$ are $64 \times 64$ images, and the proxy variable $Z$ is a three-dimensional vector, representing image scale, rotation and position in the horizontal direction (posX). The unobserved confounder is the vertical position of the image (posY). Figures 9 and 10 show true and synthesized images from the dSprite data, respectively, and Figure 11 shows the true treatments $X$ connected to Figure 10 (as emphasized in the caption to Figure 11, it is important to note that the synthesized $W$ is able to infer the proper PosY from $X$, which is inferred from the image of $X$).

## F Network Structures and Hyperparameters

In both synthetic data experiments, we use $k$-fold cross-validation ($k = 5$) to select the learning rate from the range $[10^{-3}, 10^{-4}, 10^{-5}]$. The learning rate is the same for all model components, but one can try to make the learning rate different across different components. We can also use $k$-fold cross-validation for synthetic data and the validation split for Framingham data to determine the weights for each autoencoder loss accordingly. Once the hyperparameters are selected, the model is trained with a train/validation split of $0.8/0.2$. Early stopping is implemented using the validation loss of the causal bridge $\mathcal{L}_{\theta_Y}$ only, even for the autoencoder version of the model. In Framingham experiments, we use the validation split provided in the dataset to measure the concordance index (C-Index) and select the batch size from the range $[384, 512, 768]$, keeping the learning rate and weight decay fixed at $10^{-1}$ and $10^{-4}$, respectively. The test split in Framingham is used only to

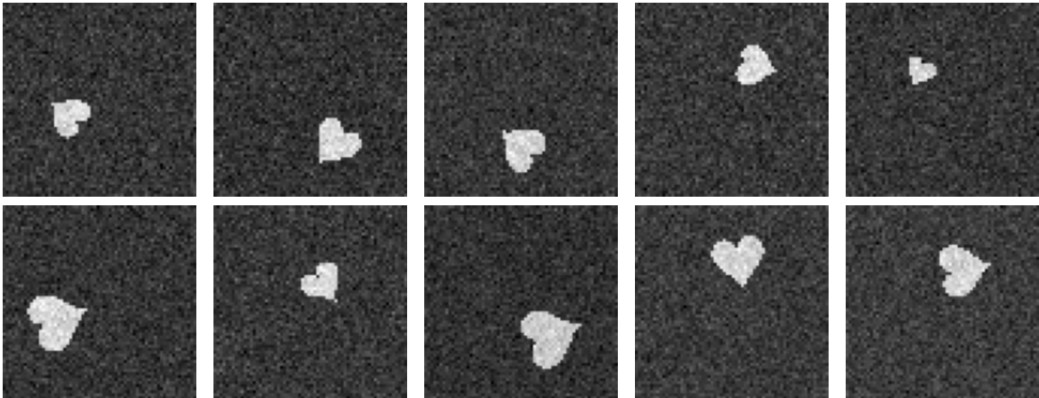

Figure 11: True treatments $X$ associated with the generative model used to draw from $p(W|Z = z, X = x)$ in Figure 10. Note that the generator is given $Z = $ (scale, rotation, posX), but it must infer the latent $U = $ posY from $X$, which shares the $U$. Note that while the treatments here have different $Z$ than that used for $W$ ($W$ is here independent of $Z$), the model is able to correctly extract posY from $X$. The subfigures here correspond to those in Figure 10.

evaluate the C-Index of the trained model. To ensure a fair comparison, we utilize the training and validation split to learn the baseline CoxPH models with three different weighting schemes.

The details of the architectures used for Demand, dSprite and Framingham are shown in Tables 2, 3, and 4, respectively.

### F.1 Demand Experiments

For the demand experiment, we use the AdamW optimizer with a weight decay of $10^{-5}$ and a learning rate of $10^{-4}$ for all models. The autoencoder loss $\mathcal{L}_{\theta_X}$ is weighted with $\mathtt{w_x} = 1$, and $\mathcal{L}_{\theta_Z}$ is weighted with $\mathtt{w_z} = 1$.

### F.2 dSprite Experiments

For the dSprite experiment, we use the Adam with learning rate of $10^{-3}$ for all models. The autoencoder loss $\mathcal{L}_{\theta_X}$ is weighted with $\mathtt{w_x} = 100$, and $\mathcal{L}_{\theta_Z}$ is weighted with $\mathtt{w_z} = 0.01$.

### F.3 Framingham Experiments

For the Framingham dataset, the model structures are all linear for the bridges. We have shown $\theta_x = (\phi_x, \phi_z)$ for ease of explanation in Table 4 separately. To model $p(W|X, Z)$, we utilize a simple MLP-based conditional diffusion probabilistic model (DDPM) [20]. To train the conditional DDPM, we use the AdamW optimizer with a weight decay of $10^{-2}$, a learning rate of $10^{-3}$ and the number of diffusion timesteps is 1000. For the bridge and autoencoder modules, we use the AdamW optimizer with a weight decay of $10^{-3}$ and a learning rate of $10^{-1}$. The autoencoder loss $\mathcal{L}_{\theta_X}$ is weighted with $\mathtt{w_x} = 0.5$, and $\mathcal{L}_{\theta_Z}$ is weighted with $\mathtt{w_z} = 0.1$.

## G CoxPH Loss with Weighting

For learning the Cox proportional hazards (CoxPH) model with various weighting schemes, we maximize the propensity-weighted partial likelihood [9, 42, 46] to obtain the parameters, $\gamma$:

$$\mathcal{L}(\gamma) = \prod_{m:e_m=1} \left( \frac{\exp(x_m\gamma)}{\sum_{n:y_n \geq y_m} \omega_n \cdot \exp(x_n\gamma)} \right)^{\omega_m} \tag{69}$$

Here, for $n^{\text{th}}$ example, $y_n$ denotes the observed time $y_n = \min(t_n, c_n)$, $t_n$ is the time at which the event of interest occurs and $c_n$ is the follow-up time. If $y_n = t_n < c_n$, it is said that the event of interest is observed and $e_n = 1$, otherwise, $y_n = c_n < t_n$, $e_n = 0$ and the event is right-censored.

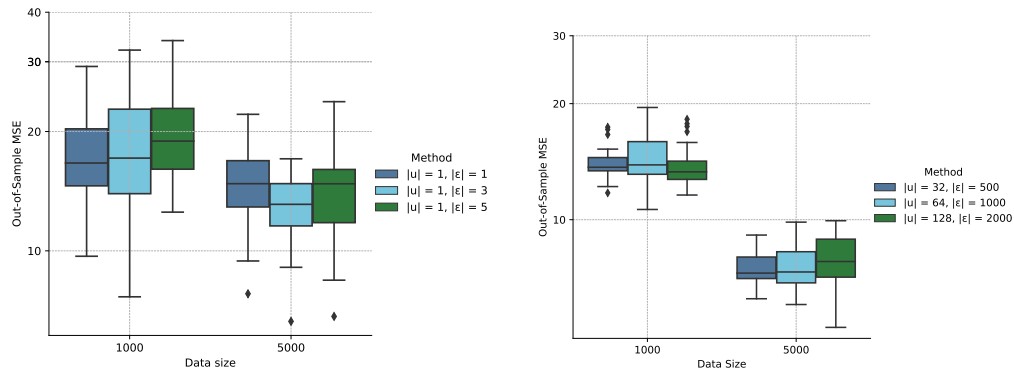

Figure 12: Ablation results with the CB + AE model when varying the dimensionality of (Left) $\epsilon$ on the **Demand** dataset and (Right) $U$ and $\epsilon$ on the **dSprite** dataset.

Table 1: Out-of-sample MSE results for **Demand** and **dSprite** data. Figures are medians in interquartile ranges (Q1 and Q3). These results match those in Figure 3, but also include results for CEVAE and KAP.

| | | Demand | dSprite |
|---|---|---|---|
| Method | Sample Size | Median (Q1, Q3) | Median (Q1, Q3) |
| CEVAE | 1000 | 360.05 (205.15, 602.59) | 52.92 (51.45, 53.29) |
| NMMR-V | 1000 | 23.07 (17.27, 28.12) | 24.48 (21.32, 25.73) |
| NMMR-U | 1000 | 21.6 (20.41, 25.38) | 22.22 (18.37, 25.57) |
| KAP | 1000 | 109.86 (79.01, 176.86) | 22.4 (21.75, 23.03) |
| DFPV | 1000 | 43.93 (38.38, 48.34) | 22.08 (19.96, 24.38) |
| DFPV (our generator) | 1000 | 34.15 (29.65, 40.88) | 15.25 (13.97, 18.59) |
| CB (our method) | 1000 | 20.5 (17.52, 24.78) | 14.06 (13.21, 14.94) |
| CB + AE (our method) | 1000 | **16.66 (14.59, 20.32)** | **13.68 (13.39, 14.49)** |
| CEVAE | 5000 | 262.24 (161.96, 438.63) | 53.89 (53.07, 54.18) |
| NMMR-V | 5000 | 31.03 (24.07, 41.42) | 11.62 (8.77, 14.44) |
| NMMR-U | 5000 | 17.12 (13.19, 24.1) | 19.11 (18.11, 20.3) |
| KAP | 5000 | 153.19 (96.4, 221.03) | 26.53 (25.85, 27.1) |
| DFPV | 5000 | 43.35 (32.65, 53.2) | 16.33 (15.23, 17.44) |
| DFPV (our generator) | 5000 | 27.84 (24.73, 30.71) | 8.79 (8.2, 9.49) |
| CB (our method) | 5000 | 15.84 (14, 17.28) | 7.45 (6.92, 7.96) |
| CB + AE (our method) | 5000 | **14.77 (12.9, 16.89)** | **7.27 (7.04, 8)** |

Thus, the weighting schemes estimate weights for each subject using the entire dataset and use these weights to fit a CoxPH model. The unknown propensity score $s_n = P(X = 1|W = w_n, Z = z_n)$ is modeled using a linear logistic regression model: $\hat{s}_n = \sigma(W = w_n, Z = z_n)$. The weight for each $m^{\text{th}}$ example is computed as follows:

- CoxPH-OW: overlapping weights,

$$\omega_m = x_i \cdot (1 - \hat{s}_m) + (1 - x_m) \cdot \hat{s}_m,$$

- CoxPH-IPW: inverse probability weighting,

$$\omega_m = \frac{x_m}{\hat{s}_m} + \frac{1 - x_m}{1 - \hat{s}_m},$$

- CoxPH-Uniform: uniform weights (standard RCT uniform assumption).

## H   Additional Results for Demand and dSprite

Table 1 present all the results for Demand and dSprite like in Figure 3, but also including CEVAE and KAP.

## I  Ablation Study

We explore the impact of the dimensionality of the latent $U$ and noise model for the shared encoder $h_{\theta_U}(W, x, \epsilon)$. For the Demand dataset we consider $|U| = 1$ and $|\epsilon| = \{1, 3, 5\}$, where $|U|$ indicates the cardinality (dimensionality) of $U$. For the dSprite dataset, we consider $|U| = \{32, 64, 128\}$ and $|\epsilon| = \{500, 1000, 2000\}$. Note that in the main results shown in Figure 3, $|U| = |\epsilon| = 1$, and $|U| = 32$ and $|\epsilon| = 500$, for the Demand and dSprites datasets, respectively. All other parameters of the model are fixed and consistent with those used for the main results, as shown in Appendix F. The optimal range for the dimensions of $U$ and $\epsilon$ can be determined by domain knowledge or cross-validation. For instance, the dimensionalities of $W$ in Demand and dSprite are $|W| = 1$ and $|W| = 4096$, respectively, although the latter is contained on a lower dimensional manifold, which explain the differences in $U$ above. The difference between $|U|$ and $|\epsilon|$ for the dSprite dataset is necessary because smaller values of $|\epsilon|$ tend to cause the learning to ignore the variation of $h_{\theta_U}(W, x, \epsilon)$, which can cause overfitting issues.

The ablation results for Demand and dSprite shown in Figure 12 indicate that the proposed CB + AE model is fairly insensitive to the dimensionality of the latent $U$ and $\epsilon$. These are a subset of all ablation studies we have considered, consistent with our experience that the proposed models train well and are not particularly sensitive to "reasonable" parameter settings.

## J  Bridge Generalization from Proposition 2

This document describes the implementation and analysis of a Structural Equation Modeling (SEM) experiment using Bayesian methods. The experiment involves estimating latent variables from observed data while accounting for confounding factors.

The observed variables are generated from the SEM in (32)-(35) with all structural coefficients $\{\alpha_{ZU}, \alpha_{WU}, \alpha_{XZ}, \alpha_{XU}, \alpha_{YX}, \alpha_{YW}, \alpha_{YU}\} = 1$, treatment and outcome variances $\{\sigma_Y, \sigma_X\} = 1$, and the latent confounder is set to $U \sim \text{Uniform}(0, 10)$. We generate $n = 100$ observations of $(x, z, w)$, Using these observations we can train a model $p(W|x, z)$, after the model is trained, we can sample $J = 50$ $W$ using $p(W|x, z)$ for every $\{x, z\}$, Using these generated $W$ we can then use MCMC to get $M = 100$ samples of $U$ from $p(U|w, x, z)$ and $p(U|w, x)$ for every $\{w, x, z\}$.

The details of experiment steps are the follows:

**Sampling $W$** We model $p(W \mid x, z)$ as a Gaussian with parameters (mean and variance) given by a neural network with inputs $\{x, z\}$. With samples $(w_i, x_i, z_i)$, we train a network to generate samples from $p(W \mid X, Z)$ using maximum likelihood estimation with gradient descent. Then for each $(x_i, z_i)$ we draw

$$w_{i,j} \sim p_\phi(W \mid x_i, z_i), \quad j = 1, \ldots, J, \tag{70}$$

where $J = 50$.

**Sampling $U$** We compile two NUTS [21] samplers in PyMC [45]: one targeting $p(U|W, X)$ and one targeting $p(U|W, X, Z)$. For each proxy sample $w_{i,j}$ we generate

$$u_{i,j,n}^{(1)} \sim p(U|w_{i,j}, x_i), \qquad u_{i,j,n}^{(2)} \sim p(U|w_{i,j}, x_i, z_i), \qquad n = 1, \ldots, M. \tag{71}$$

where $M = 100$.

**Learning the Bridge Function** Define a feedforward network $g_\theta(u, w, x)$. We approximate the bridge using Proposition 2 as

$$b_\theta(W, x) \approx \frac{1}{M} \sum_{n=1}^{M} g_\theta\left(u_{i,j,n}^{(1)}, w_{i,j}, x_i\right). \tag{72}$$

Then we learn $\theta$ in $b_\theta(W, x)$ by minimizing

$$\hat{\theta} = \arg\min_\theta \sum_{i=1}^{n} \left(\mathbb{E}[Y \mid x_i, z_i] - \mathbb{E}[b_\theta(W, x) \mid x_i, z_i]\right)^2. \tag{73}$$

**Evaluation** Finally, we compute the $\frac{\eta_i}{|\mathbb{E}[Y|x_i, z_i]|}$ on the causal effect per data point as

$$\frac{\eta_i}{|\mathbb{E}[Y|x_i, z_i]|} = \frac{\left|\mathbb{E}[Y|x_i, z_i] - \mathbb{E}[b_\theta(W, x)|x_i, z_i]\right|}{\left|\mathbb{E}[Y|x_i, z_i]\right|}, \tag{74}$$

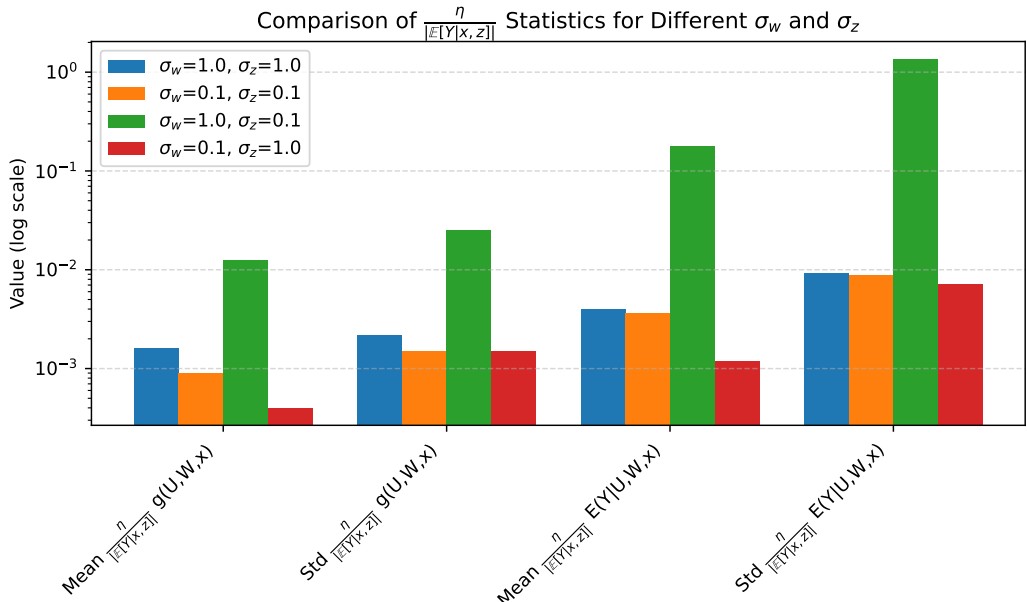

Figure 13: The statistics (mean and standard deviation) of $\frac{\eta}{|\mathbb{E}[Y|x,z]|}$ for learning the bridge function compared to using known $\mathbb{E}(Y|U, W, X)$.

and report the empirical mean and standard deviation of $\{ \frac{\eta_i}{|\mathbb{E}[Y|x_i,z_i]|} \}_{i=1}^n$ for $n = 100$. Note that this mean is an approximation to the mean of $r(\eta)$.

**Results** $\mathbb{E}[Y|x_i, z_i]$ is calculated using samples from $p(U|w,x,z)$, $p(W|x,z)$ and observed $(x, z)$, using the true expectation $\mathbb{E}(Y|U, W, X) = x + W + U$, where $\mathbb{E}[b_\theta(W, x)|x_i, z_i]$ is calculated using samples from $p(U|w,x)$, $p(W|x,z)$ and observed $(x, z)$ using:

- The true expectation $\mathbb{E}(Y|U, W, X) = x + W + U$.
- The learned function $g(U, W, x)$.

Results for the mean and standard deviation of $\{ \frac{\eta_i}{|\mathbb{E}[Y|x_i,z_i]|} \}_{i=1}^n$ in Figure 13.

## K  Additional Survival Analysis Results

For the Framingham dataset, we plot the KM curves, which provide the survival probability estimates without considering the confounding factors. Figure 14 shows the Kaplan-Meier (KM) survival curves for the two groups with treatment $X = 1$ and treatment $X = 0$. Notably, the KM curves indicate that the group of patients that received the treatment ($X = 1$) has decreased survival probabilities compared to the group that received the treatment ($X = 0$), even though previous large-scale longitudinal RCT trials indicated a hazard ratio (HR) of 0.75. This emphasizes the necessity of modeling the Framingham survival analysis using a framework that precisely captures the causal relationship.

For the Framingham dataset, we report the concordance index (C-Index) to show the effectiveness of our proposed approach to correctly rank the survival times. The interpretation of C-Index is similar to the area under the ROC curve (AUC), but it is a generalization for censored data. CI ranges from 0 to 1, such that a score of 1 implies a perfect ranking, 0.5 implies random predictions and $< 0.5$ indicates performance worse than random.

Figure 15 shows the box plots for the hazard ratios and concordance index (CI) on the test split over 30 runs. Our proposed framework of the causal bridge along with the autoencoder (CB + AE) outperforms only using the causal bridge (AE) in terms of both the hazard ratio and concordance

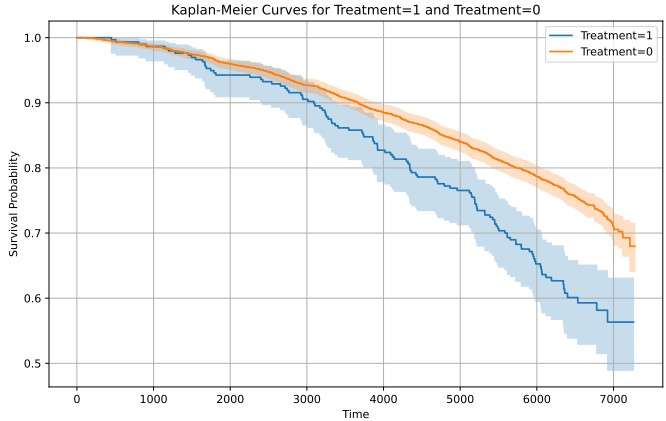

Figure 14: Framingham dataset: Kaplan Meier Curves.

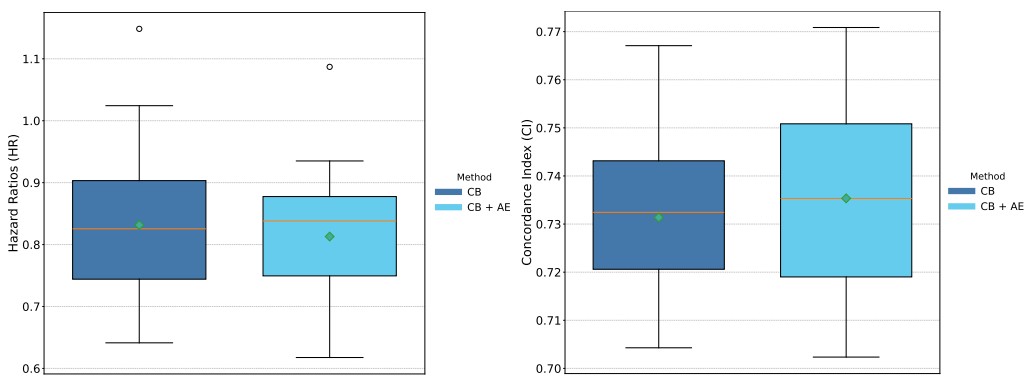

Figure 15: Framingham dataset: (Left) Hazard ratio and (Right) Concordance Index (CI).

index, demonstrating that using AutoEncoder helps not only in learning the causal relationship but also improves the predictive capability of the model.

Table 2: Models for the Demand Experiment.

| Model | $p(W|Z=z, X=x)$ |
|---|---|
| 1 | Input$(z,x)$ |
| 2 | FC$(3,32)$, ReLU |
| 3 | FC$(32,64)$, ReLU |
| 4 | FC$(64,16)$, ReLU |
| 5 | Mean: FC$(16,1)$ |
| 6 | Std: FC$(16,1)$, Softplus |

| Model | $h(w,x,\epsilon)$ |
|---|---|
| 1 | Input$(w,\epsilon)$ |
| 2 | FC$(3,32)$, ReLU |
| 3 | FC$(32,64)$, ReLU |
| 4 | FC$(64,16)$, ReLU |
| 5 | FC$(16,1)$ |

| Model | $\theta_y$ |
|---|---|
| 1 | Input$(x,u,w)$ |
| 2 | FC$(3,32)$, ReLU |
| 3 | FC$(32,64)$, ReLU |
| 4 | FC$(64,16)$, ReLU |
| 5 | FC$(16,1)$ |

| Model | $\phi_X$ |
|---|---|
| 1 | Input$(x)$ |
| 2 | FC$(1,32)$, ReLU |
| 3 | FC$(32,64)$, ReLU |
| 4 | FC$(64,16)$, ReLU |
| 5 | FC$(16,1)$ |

| Model | $\theta_x$ |
|---|---|
| 1 | Input$(z,u)$ |
| 2 | FC$(3,32)$, ReLU |
| 3 | FC$(32,64)$, ReLU |
| 4 | FC$(64,16)$, ReLU |
| 5 | FC$(16,1)$ |

| Model | $\theta_z$ |
|---|---|
| 1 | Input$(u)$ |
| 2 | FC$(1,32)$, ReLU |
| 3 | FC$(32,32)$, ReLU |
| 5 | FC$(32,2)$ |

Table 3: Models for the dSprite Experiment.

| Model | $p(W|Z=z, X=x)$ G |
|---|---|
| 1 | Input$(x,z,\text{noise})$ |
| 2 | FC$(4299,2048)$, ReLU |
| 3 | FC$(2048,4096)$ |

| Model | $p(W|Z=z, X=x)$ D |
|---|---|
| 1 | Input$(w,x,z)$ |
| 2 | FC$(8195,4096)$, ReLU |
| 3 | FC$(4096,1)$, Sigmoid |

| Model | $h(W,x,\epsilon)$ |
|---|---|
| 1 | Input$(w,x,\epsilon)$ |
| 2 | FC$(8692,1024)$, ReLU |
| 3 | FC$(1024,256)$, ReLU |
| 4 | FC$(256,64)$, ReLU |
| 5 | FC$(64,32)$ |

| Model | $\phi_X$ |
|---|---|
| 1 | Input$(x)$ |
| 2 | FC$(4096,64)$, ReLU |
| 3 | FC$(64,32)$ |

| Model | $\theta_y$ |
|---|---|
| 1 | Input$(x,u,w)$ |
| 2 | FC$(4160,1024)$, ReLU |
| 3 | FC$(1024,256)$, ReLU |
| 4 | FC$(256,1)$ |

| Model | $\theta_x$ |
|---|---|
| 1 | Input$(z,u)$ |
| 2 | FC$(35,64)$, ReLU |
| 3 | FC$(64,128)$, ReLU |
| 4 | FC$(128,256)$, ReLU |
| 5 | FC$(256,4096)$ |

| Model | $\theta_z$ |
|---|---|
| 1 | Input$(u)$ |
| 2 | FC$(32,512)$, ReLU |
| 3 | FC$(512,3)$ |

Table 4: Models for the Framingham Experiment.

| | Embedding Layer |
|---|---|
| 1 | Input$(x, z)$ |
| 2 | FC(17, 64), Linear |

| | Time Step Embedding Layers |
|---|---|
| 1 | FC(64, 64), Linear |
| 2 | SiLU Activation |
| 3 | FC(64, 64), Linear |

| | Projection Layer |
|---|---|
| 1 | FC(16, 64), Linear |

| | DDPM Structure |
|---|---|
| 1 | FC(64, 256), ReLU, Dropout(0.1) |
| 2 | FC(256, 512), ReLU, Dropout(0.1) |
| 3 | FC(512, 256), ReLU, Dropout(0.1) |
| 4 | FC(256, 16), Linear |

| **Model** | $h(W, X, \epsilon)$ |
|---|---|
| 1 | Input$(w, x, \epsilon)$ |
| 2 | FC(18, 1) |

| **Model** | $\theta_y$ |
|---|---|
| 1 | Input$(x, u, w)$ |
| 2 | FC(18, 1) |

| **Model** | $\phi_z$ (part of $\theta_x$) |
|---|---|
| 1 | Input$(z)$ |
| 2 | FC(16, 16) |

| **Model** | $\phi_x$ (part of $\theta_x$) |
|---|---|
| 1 | Input$(\phi(z), u)$ |
| 2 | FC(17, 1) |

| **Model** | $\theta_z$ |
|---|---|
| 1 | Input$(u)$ |
| 2 | FC(1, 16) |

