# OpenReview forum: "Coupling Generative Modeling and an Autoencoder with the Causal Bridge"
_NeurIPS.cc/2025/Conference — NeurIPS 2025 poster_

### Official Review · Reviewer_rsP2 · 2025-06-29

**Clarity:** 3
**Significance:** 2
**Originality:** 1
**Rating:** 4
**Confidence:** 5

**Summary:**

This paper derives theoretical bounds on the average error in treatment effect estimation under proximal causal inference when the causal bridge (CB) assumption is violated. The authors also propose an autoencoder-based architecture for estimating the causal bridge function, providing both theoretical guarantees and a practical implementation framework.

**Questions:**

I might increase or decrease my score based on authors' responses
- What motivated the choice of autoencoders over diffusion models for generating samples from the conditional distribution, particularly given diffusion models' demonstrated effectiveness in generative tasks?
- Could you include comparisons with the missing baselines identified in the weakness section (CAVAE, Kompa et al. 2022, and Ying et al. 2022) to provide a more comprehensive experimental evaluation?

**Ethical Concerns:**

["NO or VERY MINOR ethics concerns only"]

**Final Justification:**

The authors responses addressed my concerns and questions. As such I increased the score.

**Limitations:**

The authors fail to acknowledge that their experimental evaluation omits several important baselines from recent literature, and do not discuss why they chose auto-encoders instead of diffusion models.

**Quality:**

2

**Strengths And Weaknesses:**

*Strengths*:
- The paper extends existing theory for proximal causal inference by providing theoretical analysis for the scenario where the bridge function independence assumption (relative to the proxy variable) is violated.
- The experimental evaluation includes both synthetic datasets and real-world data with ground truth established through randomized controlled trials, providing comprehensive empirical validation.

*Weaknesses*:
- The paper fails to discuss or compare against recent neural network approaches for unmeasured confounding, including key works such as Deep Learning Methods for Proximal Inference via Maximum Moment Restriction (Kompa et al., 2022), Causal Effect Estimation using Identifiable Variational AutoEncoder with Latent Confounders and Post-Treatment Variables (Xie et al., 2024), and Density Ratio-based Proxy Causal Learning Without Density Ratios (Bozkurt et al., 2025).
- The synthetic data experiments omit important baselines: (1) CAVAE: the authors' claim that results are not numerically stable does not justify excluding this established method, and (2) the approach from Kompa et al. (2022).
- The real-world data evaluation lacks comparison with relevant baselines, particularly Ying et al. (2022).
- The authors present their method's proportional hazard assumptions as an advantage over Ying et al. (2022) in survival outcomes, but fail to acknowledge that requiring this assumption could be viewed as a limitation of their own approach.
- The paper provides no justification for choosing autoencoders over diffusion models for sampling, despite diffusion models' growing prominence in generative modeling

---

> ### Author Rebuttal · Authors · 2025-07-30
>
> We would like to thank the reviewer for their constructive feedback. Below we address the three concerns raised by the reviewer.
>
> 1. Thanks for pointing us to the three related papers. Below we show results for Kompa et al., 2022 and Bozkurt et al., 2025, showing that the proposed approaches (CB and CB + AE, namely with and without the autoencoder formulation, respectively) outperform both of their settings NNMR-U and NNMR-V, and Kernel Alternative Proxy, respectively. In the table below, Q1 and Q3 are the first and third quartiles presented along with the median out-of-sample MSE results. The other paper will be discussed in the revision; however, it is not considered in the experiments because Xie et al., 2024 is for a different setting that requires post-treatment variables.
>
> | Method |	Sample Size |	Demand Median (Q1, Q3) |	dSprite Median (Q1, Q3) |
> | :--- | :----: | :----: | :----: |
> | CEVAE |	1000 |	360.05 (205.15, 602.59) |	52.92 (51.45, 53.29) |
> | NMMR-V |	1000 |	23.07 (17.27, 28.12) |	24.48 (21.32, 25.73) |
> | NMMR-U |	1000 |	21.6 (20.41, 25.38) |	22.22 (18.37, 25.57) |
> | Kernel Alternative Proxy |	1000 |	109.86 (79.01, 176.86) |	22.4 (21.75, 23.03) |
> | DFPV |	1000 |	43.93 (38.38, 48.34) |	22.08 (19.96, 24.38) |
> | DFPV $\sim p(W\|x,z)$ (our generator) |	1000 |	34.15 (29.65, 40.88) |	15.25 (13.97, 18.59) |
> | CB (our method) |	1000 |	20.5 (17.52, 24.78) |	14.06 (13.21, 14.94) |
> | CB + AE (our method) |	1000 |	**16.66 (14.59, 20.32)** |	**13.68 (13.39, 14.49)** |
> | CEVAE |	5000 |	262.24 (161.96, 438.63) |	53.89 (53.07, 54.18) |
> | NMMR-V |	5000 |	31.03 (24.07, 41.42) |	11.62 (8.77, 14.44) |
> | NMMR-U |	5000 |	17.12 (13.19, 24.1) |	19.11 (18.11, 20.3) |
> | Kernel Alternative Proxy |	5000 |	153.19 (96.4, 221.03) |	26.53 (25.85, 27.1) |
> | DFPV |	5000 |	43.35 (32.65, 53.2) |	16.33 (15.23, 17.44) |
> | DFPV $\sim p(W\|x,z)$ (our generator) |	5000 |	27.84 (24.73, 30.71) |	8.79 (8.2, 9.49) |
> | CB (our method) |	5000 |	15.84 (14, 17.28) |	7.45 (6.92, 7.96) |
> | CB + AE (our method) |	5000 |	**14.77 (12.9, 16.89)** |	**7.27 (7.04, 8)** |
>
> 2. Above we show results for CEVAE, NNMR-U, NNMR-V (Kompa et al., 2022) and Kernel Alternative Proxy (Bozkurt et al., 2025), from which we see that the proposed approach outperforms the alternatives in both datasets (Demand and dSprite) and dataset sizes (1000 and 5000). These will be shown graphically in the revision as in Figure 3.
>
> 3. We produced results using the method in Ying et al. 2022 using the same data settings in our experiment with the Framingham dataset shown in Figure 3 (Right). We use their formulation for the hazard function (described below as $h(t,W,X,V)$) and estimate hazard ratios (HRs) for three distinct event quantile time points $t=(0.25,0.5,0.75)$ which correspond to event horizons at 1924, 3709, and 5494 days, respectively, consistent with the experiments in Ying et al. 2022. Note that unlike in our approach, they estimate causal effects for specific points in time, while we consider a time-independent causal estimate. Results below indicate that the approach in Ying et al. 2022 fails to capture the correct direction of the hazard ratio (HR$< 1$) and magnitude of the ground-truth HR=0.75. Summaries using the method of Ying et al. 2022 (mean and standard deviation for the HR) are calculated over 30 runs:
>
> | **t**         | **0.25** | **0.5**  | **0.75** |
> |:--------------|---------:|---------:|---------:|
> | **Mean**      | 1.0454   | 1.1414   | 1.2103   |
> | **Standard Deviation** | 0.0011   | 0.0077   | 0.0007   |
>
> 4. We understand that the proportional hazard assumption can be seen as a limitation; however, in Ying et al., 2022, they consider a very specific formulation for the hazard function that can be similarly perceived as a limitation. Specifically, they use $h(t,W,X,V) = exp( - b_0 t - b_1 t^2 - b_a t X - b_w t W - b_v V )$, *i.e.*, an exponentiated linear function of time ($t$), time $\times$ time ($t^2$), time $\times$ treatment ($t X$), time $\times$ outcome proxy ($t W$) and additional covariates denoted as $V$. In the revision, we will acknowledge the limitations of both approaches, *i.e.*, using a proportional hazard assumption *vs.* choosing a specific form of the hazard function a priori.
>
> 4. We understand the confusion about choosing the autoencoder for sampling, which we clarify as follows. We are not using the autoencoding formulation for sampling, but to learn functions $g_{\theta_Y}$, $g_{\theta_Z}$, $g_{\theta_X}$, and $h_{\theta_U}$. Sampling is done through a separate model trained to sample from $w \sim p(W|x,z)$, which is not implemented with an autoencoder. In fact, we use a conditional GAN and a conditional diffusion probabilistic model for the dSprite and Framingham experiments as described in Appendices E.5 and F.3, respectively. The Demand data is very simple; thus, a simple MLP modeling the (Gaussian) mean and variance of W is sufficient as described in Appendix E.4. In answer to the question from the reviewer, we are effectively using a diffusion model when there is a need for sampling.

---

> > ### Comment · Reviewer_rsP2 · 2025-08-02
> >
> > Thanks for providing the results from additional baselines and clarifications on my questions. I increased my score based on the updated results and arguments from the authors.

---

### Official Review · Reviewer_vSpr · 2025-07-02

**Clarity:** 3
**Significance:** 2
**Originality:** 3
**Rating:** 4
**Confidence:** 3

**Summary:**

This paper focuses on the effect estimation with latent variables and two additional observed proxies,  a framework commonly referred to as proximal methods. The paper derives a new error bound on the causal bridge function (more specifically,  the outcome confounding bridge function), highlighting that the proxies should serve as low-noise measurements of the latent confounders. Further, the paper proposes a generative model based on an autoencoder architecture to learn the bridge function. Experimental results demonstrate the effectiveness of the proposed approach.

**Questions:**

see above

**Ethical Concerns:**

["NO or VERY MINOR ethics concerns only"]

**Final Justification:**

During the rebuttal, all my concerns were well addressed. Thus I increase my score from 3 to 4.

**Limitations:**

see above

**Quality:**

2

**Strengths And Weaknesses:**

### strengths

1. The paper is clearly written and well-organized.

1. It proposes an autoencoder-based architecture for learning the causal bridge function within the proximal framework, which performs well empirically.



### Weaknesses / Questions

1. Corollary is not a surprising result, since $\Psi$ is a non-information-loss function. Therefore, the information loss stems solely from the exogenous noise $\epsilon_w$.  As $\sigma_{\epsilon_w} \rightarrow 0$, the proxy $W$ become equivalent to $U$, leading to $I(U;Z|W) \approx I(\Psi(W);Z|W)=I(W;Z|W)=0$.
1. The authors discuss solely the case of outcome confounding bridge function $b$ (denoted $h$ in [1]). Is it possible to drive similar result for treatment  confounding bridge function $q$​ [1]?
1. Relatedly, is it possible to learn the treatment bridge function $q$ using the proposed autoencoder architecture?
1. It would be better to clarify why we should access $D_1$. What is the implication of using a separate dataset versus the same dataset?
1. The experimental evaluation could be strengthened by comparing the proposed method with more state-of-the-art proximal baselines.
1.  The paper discusses how the additional loss terms in Equation (13) improve the learned bridge function hhh, but an ablation study on these terms is missing.



[1] Semiparametric proximal causal inference. JASA 2024.

---

> ### Author Rebuttal · Authors · 2025-07-30
>
> We would like to thank the reviewer for their positive assessment of our paper and constructive feedback. Below we address the concerns raised by the reviewer.
>
> 1. We completely agree with the reviewer that the Corollary is not surprising, but we decided to include it because it is an interesting special case for structural equation models. To our knowledge Theorem 3 is new, and we hoped to aid understanding of it by demonstrating that it implies the expected result in the Corollary.
>
> 2. It is certainly possible to derive a similar result for the treatment confounding bridge; this is an excellent question. We have results (experimental and theoretical) for the treatment bridge, which we have obtained since we submitted to NeurIPS, however, we consider them a separate contribution. Nevertheless, we will discuss the treatment bridge as a natural extension that will be addressed in subsequent work. Additionally, please note that the treatment bridge is most useful for binary treatments (interventions), and most of our experiments considered real treatments.
>
> 3. The reviewer is also correct in that it is possible to learn the treatment bridge in an autoencoder setting, so both the treatment and outcome bridge share statistical strength when learning the function that is used to implicitly sample from $u_j$ using $u_j = h(w_j,x,\epsilon_j)$, which for the treatment bridge will be a function of $z_j$, matched in distribution to that using $w_j$ via an integral probability measure (IPM) or minimum mean discrepancy (MMD).
>
> 4. We should clarify that $D_1$ is not necessary; however, it is done to weaken the requirement of having a common dataset for which ($W$,$Z$,$X$,$Y$) are available. This distinction is especially useful in situations where the dataset for which the outcome is available $D_2=\{(X,Z,Y)\}$ is not the same or is (much) smaller than $D_1=\{(X,Z,W)\}$. We will clarify it in the revision to avoid confusion.
>
> 5. Following your suggestion and that of **Reviewer rsP2** we have produced results for CEVAE, NNMR (Kompa et al., 2022) and Kernel Alternative Proxy (Bozkurt et al., 2025) reproduced below, where Q1 and Q3 are the first and third quartiles presented along with the median out-of-sample MSE results. In the revision these will be presented graphically as in Figure 3. We see that our methods (CB and CB + AE, namely with and without the autoencoder formulation, respectively) outperform all the baselines considered.
>
> | Method |	Sample Size |	Demand Median (Q1, Q3) |	dSprite Median (Q1, Q3) |
> | :--- | :----: | :----: | :----: |
> | CEVAE |	1000 |	360.05 (205.15, 602.59) |	52.92 (51.45, 53.29) |
> | NMMR-V |	1000 |	23.07 (17.27, 28.12) |	24.48 (21.32, 25.73) |
> | NMMR-U |	1000 |	21.6 (20.41, 25.38) |	22.22 (18.37, 25.57) |
> | Kernel Alternative Proxy |	1000 |	109.86 (79.01, 176.86) |	22.4 (21.75, 23.03) |
> | DFPV |	1000 |	43.93 (38.38, 48.34) |	22.08 (19.96, 24.38) |
> | DFPV $\sim p(W\|x,z)$ (our generator) |	1000 |	34.15 (29.65, 40.88) |	15.25 (13.97, 18.59) |
> | CB (our method) |	1000 |	20.5 (17.52, 24.78) |	14.06 (13.21, 14.94) |
> | CB + AE (our method) |	1000 |	**16.66 (14.59, 20.32)** |	**13.68 (13.39, 14.49)** |
> | CEVAE |	5000 |	262.24 (161.96, 438.63) |	53.89 (53.07, 54.18) |
> | NMMR-V |	5000 |	31.03 (24.07, 41.42) |	11.62 (8.77, 14.44) |
> | NMMR-U |	5000 |	17.12 (13.19, 24.1) |	19.11 (18.11, 20.3) |
> | Kernel Alternative Proxy |	5000 |	153.19 (96.4, 221.03) |	26.53 (25.85, 27.1) |
> | DFPV |	5000 |	43.35 (32.65, 53.2) |	16.33 (15.23, 17.44) |
> | DFPV $\sim p(W\|x,z)$ (our generator) |	5000 |	27.84 (24.73, 30.71) |	8.79 (8.2, 9.49) |
> | CB (our method) |	5000 |	15.84 (14, 17.28) |	7.45 (6.92, 7.96) |
> | CB + AE (our method) |	5000 |	**14.77 (12.9, 16.89)** |	**7.27 (7.04, 8)** |
>
> 6. We apologize for the confusion regarding equation (13). We do have an ablation study in the sense that we present results both using (12), *i.e.*, just the bridge (CB in Figure 3), and (12) + (13), *i.e.*, the proposed autoencoding setup (CB + AE in Figure 3). These are reproduced above for convenience. Moreover, the weights of the three loss terms ($L_{\theta_Y}$, $L_{\theta_X}$, $L_{\theta_Z}$) are also optimized as described in Appendix F. Moreover, we do not present ablation results only using $L_{\theta_X}$ and $L_{\theta_Z}$ because the autoencoding setup is intended to model the joint distribution $p(X,Z|U)$ following the graphical model in Figure 1.

---

> > ### Comment · Reviewer_vSpr · 2025-08-04
> > **Thanks**
> >
> > I appreciate the authors' detailed responses, especially the additional experimental results, well supporting their claims.
> >
> > I modified my score correspondingly,

---

### Official Review · Reviewer_5YZq · 2025-07-03

**Clarity:** 3
**Significance:** 3
**Originality:** 3
**Rating:** 4
**Confidence:** 5

**Summary:**

The paper presents theoretical insights into the causal bridge method, a causal inference approach based on proxy variables, designed to estimate the causal effect of a treatment variable on an outcome in the presence of unobserved confounders. Building on these theoretical analyses, the paper proposes an architecture aimed at improving the quality of causal bridge estimates.

**Questions:**

Please refer to Strengths And Weaknesses.

**Ethical Concerns:**

["NO or VERY MINOR ethics concerns only"]

**Final Justification:**

I believe the paper addresses an interesting problem and proposes both theoretical analysis an a well-justified methodology. However, the concerns regarding the clarity of the paper remains.Overall, I'd like to keep my positive score.

**Limitations:**

Yes.

**Quality:**

3

**Strengths And Weaknesses:**

Strengths:

1. The paper is well-motivated and addresses an important problem. Causal effect estimation in the presence of unobserved confounding is a challenging yet critical issue, and proposing a framework to handle this is valuable.
2. The paper is interesting, particularly the theoretical analysis and how it is used to justify the proposed method. The comparison with CEVAE is also compelling.
3. The paper is well-structured and easy to follow and the proposed method demonstrates good performance, outperforming the baselines.


Weaknesses:

1. Some contributions of the paper appear incremental. In particular, much of Section 2, e.g., including Propositions 1 and 2, are direct derivations from Equation (1). Additionally, some findings, such as W and Z are noisy mappings of U have already been discussed in prior work [20].
2. Some of the language in the paper needs to be more precise. For example, the introduction should explain more clearly the advantages of proxy variable methods, especially causal bridge, compared to others, such as instrumental variable approaches.
3. The paper only uses one real-world dataset in the experiments. Including more real-world datasets would help better demonstrate the performance of the proposed method. Also, given the similarities between the proposed approach and the CEVAE, the paper should also include that as baseline. Also, I highly suggest that the paper includes more ablation studies in the main paper, rather than the appendix.

---

> ### Author Rebuttal · Authors · 2025-07-30
>
> We would like to thank the reviewer for their positive assessment of our paper and constructive feedback. Below we address the three concerns raised by the reviewer.
>
> 1. Some of the contents of the paper including Propositions 1 and 2 though can be derived from (1) as the reviewer point out have not been introduced elsewhere. However, the reviewer is correct in that the assumption that $W$ and $Z$ are noisy mappings of $U$ have been discussed elsewhere. We would like to point out that beyond re-examining the assumptions underlying the causal bridge in Section 4, which includes new theoretical results in Theorem 3 and Corollary 1, we also introduce a new formulation for the causal bridge leveraging generative models for the outcome proxy ($W$), an autoencoder architecture that enables sharing of statistical strength between observed quantities ($W$, $Z$, $X$ and $Y$), an extension to survival outcomes and extensive experimental results with synthetic and artificial data demonstrating the effectiveness of our generative, autoencoding and survival approaches to model the causal bridge.
>
> 2. The suggestion about clarifying the advantages of proxy variable methods is well taken. In the revision, we will clarify the need and advantages of proxy variable methods in the unobserved confounding setting. Importantly, we will emphasize that it requires weaker assumptions than an instrumental variable approach, *i.e.*, that the instrumental variable and the unobserved confounder are independent and the latter influences the treatment but not the outcome.
>
> 3. We agree that having additional real-world datasets will make the experimental results stronger; however, it is difficult to find real-world datasets for which the ground-truth causal effects are known, such as it is for the case shown in our experiments using the Framingham dataset. This will be discussed as a limitation of the proposed approach in the Conclusions section. We did not include results for CEVAE because they are not competitive (they did not fit within the range of the vertical axes we used in our figure); however, we will include them in the revision and are presented below (in the revision these will be presented graphically as in Figure 3), where Q1 and Q3 are the first and third quartiles presented along with the median out-of-sample MSE results. We also include results from two additional competitive baselines suggested by **Reviewer rsP2**, namely NMMR (Kompa et al., 2022) and Kernel Alternative Proxy (Bozkurt et al., 2025). We see that our methods (CB and CB + AE, namely with and without the autoencoder formulation, respectively) outperform all the baselines considered. We agree with the reviewer that including the ablation studies in the main paper is desirable, and we plan to do so in the revision, considering that we will have additional space (1 page).
>
>
> | Method |	Sample Size |	Demand Median (Q1, Q3) |	dSprite Median (Q1, Q3) |
> | :--- | :----: | :----: | :----: |
> | CEVAE |	1000 |	360.05 (205.15, 602.59) |	52.92 (51.45, 53.29) |
> | NMMR-V |	1000 |	23.07 (17.27, 28.12) |	24.48 (21.32, 25.73) |
> | NMMR-U |	1000 |	21.6 (20.41, 25.38) |	22.22 (18.37, 25.57) |
> | Kernel Alternative Proxy |	1000 |	109.86 (79.01, 176.86) |	22.4 (21.75, 23.03) |
> | DFPV |	1000 |	43.93 (38.38, 48.34) |	22.08 (19.96, 24.38) |
> | DFPV $\sim p(W\|x,z)$ (our generator) |	1000 |	34.15 (29.65, 40.88) |	15.25 (13.97, 18.59) |
> | CB (our method) |	1000 |	20.5 (17.52, 24.78) |	14.06 (13.21, 14.94) |
> | CB + AE (our method) |	1000 |	**16.66 (14.59, 20.32)** |	**13.68 (13.39, 14.49)** |
> | CEVAE |	5000 |	262.24 (161.96, 438.63) |	53.89 (53.07, 54.18) |
> | NMMR-V |	5000 |	31.03 (24.07, 41.42) |	11.62 (8.77, 14.44) |
> | NMMR-U |	5000 |	17.12 (13.19, 24.1) |	19.11 (18.11, 20.3) |
> | Kernel Alternative Proxy |	5000 |	153.19 (96.4, 221.03) |	26.53 (25.85, 27.1) |
> | DFPV |	5000 |	43.35 (32.65, 53.2) |	16.33 (15.23, 17.44) |
> | DFPV $\sim p(W\|x,z)$ (our generator) |	5000 |	27.84 (24.73, 30.71) |	8.79 (8.2, 9.49) |
> | CB (our method) |	5000 |	15.84 (14, 17.28) |	7.45 (6.92, 7.96) |
> | CB + AE (our method) |	5000 |	**14.77 (12.9, 16.89)** |	**7.27 (7.04, 8)** |

---

> > ### Comment · Reviewer_5YZq · 2025-08-05
> >
> > I'd like to thank the authors for their response. I'll maintain my positive score.

---

### Official Review · Reviewer_2mQh · 2025-07-03

**Clarity:** 2
**Significance:** 3
**Originality:** 3
**Rating:** 4
**Confidence:** 3

**Summary:**

This paper addresses the challenge of estimating treatment effects under unobserved confounding by leveraging a causal bridge function within a proxy variable setting. It reformulates the bridge in a way that enables an intuitive deep learning implementation that improves the interpretability and statistical performance. The authors present theoretical conditions under which treatment effects can be identified using two proxy variables and introduce a novel set of assumptions that maintain the same inferential soundness. They further contribute a sensitivity analysis framework to assess robustness to assumption violations and demonstrate how their method enables statistical strength-sharing across variables. The framework consists of a two-step modeling pipeline: training a generative model to sample from proxy variables and then estimating treatment effects using these samples along with simulated unobserved confounders. A joint modeling strategy is employed to refine the confounder sampler. The approach is also extended to survival outcomes and is shown empirically to outperform state-of-the-art methods.

**Questions:**

Full disclosure: I have previously reviewed this paper and gave it an accepting score. In the rebuttal, my questions were answered, and my opinion of the paper remains largely unchanged.

**Ethical Concerns:**

["NO or VERY MINOR ethics concerns only"]

**Final Justification:**

I maintain my score and recommend the paper for acceptance.

**Limitations:**

Limitations are stated.

**Paper Formatting Concerns:**

No issues found.

**Quality:**

3

**Strengths And Weaknesses:**

**Strengths:**

1. The main contribution of the new bridge formulation is a novel result that cleverly allows a new practical implementation for estimating treatment effects.
2. Assumptions are very clearly stated.
3. Both the causal bridge approach and the joint objective are clearly demonstrated to lower estimate errors in out-of-sample settings in the experiments.

**Weaknesses:**

4. The paper is dense and is difficult to read. It would benefit from having figures and examples explaining the intuition of the bridge function, as well as the proven results.

---

> ### Author Rebuttal · Authors · 2025-07-30
>
> We would like to thank the reviewer for their positive assessment of our paper and general feedback. We understand that the paper is somewhat dense, however, necessarily so due to space constraints. We have done our best to balance its contents with extensive supplementary information in the Appendix containing an illustrative example of the bridge function as well as experiments using an intuitive structural equation model (SEM) in Appendices C and D. As a complement, we will include a new figure in Appendix C illustrating the SEM model in equations (27)-(30), which can further provide intuition about the bridge function.

---

> > ### Comment · Reviewer_2mQh · 2025-08-06
> >
> > I thank the authors for the rebuttal. I maintain my positive score.

---

### Note · Authors · 2025-08-13

Dear AC and Reviewers,

We are grateful to the reviewers for their positive assessment and valuable suggestions. We believe that incorporating their feedback will substantially improve our work. Below we restate our contributions and highlight the proposed changes as detailed in the rebuttal.

Contributions:
- **Theory:** We present a new bound (Theorem 3) on the average error of the treatment effect when the assumption that the causal bridge is independent of the treatment proxy is violated. Also a special case of the bound (Corollary 1) when the outcome proxy is assumed to be a noisy nonlinear mapping of the unobserved confounder and illustrate the new bound using data from a structural equation model (Appendix C).
- **Methods:** We introduce a new formulation for the causal bridge using generative models (GANs and diffusion models) to sample from the conditional distribution of the outcome proxy given the treatment and treatment proxy (Sec 5). We propose an autoencoder (AE) that enables the sharing of statistical strength between observed variables (Sec 6), which empirically improves causal effect estimates. Also, we extend the methodology to survival outcomes (Sec 6).
- **Experiments:** Results on synthetic and real-world data demonstrate the effectiveness of our approach including comparisons to simplified versions of our method, state-of-the-art baselines and for the real-world dataset, comparison of causal effects estimates to a randomized clinical trial. In the appendix we present detailed results for all experiments including an ablation study.

Summary of changes:
- New results for the synthetic data using three baselines (NMMR, Kernel Alternative Proxy and CEVAE) and for the real-world survival data using Ying et al. (2022), all further demonstrating the effectiveness of our method.
- We will clarify the advantages of proxy variable methods in the unobserved confounding setting; that Corollary 1 is an special case of our general Theorem 3; that results in Fig 1 for CB are an ablation of the proposed approach without the AE (CB + AE); the differences between our formulation for survival outcomes and that of Ying et al. (2022); and that in our approach, the generative model is only used to sample from the outcome proxy and not to optimize the bridge function.
- We will discuss that Xie et al. (2024) is not considered because it requires post-treatment variables and that extensions to the treatment proxy setting will be considered in future work.

---

### Decision · Program_Chairs · 2025-09-17

**Decision:**

Accept (poster)

**Comment:**

The paper provides a new error bound for the causal bridge under violated assumptions, introduces a generative + autoencoder formulation that improves estimation, extends to survival outcomes, and shows empirical gains over strong baselines on synthetic and real-world data. In addition, the authors have effectively addressed the reviewers’ concerns. Therefore, I believe this paper is suitable for publication.